# Proximal protein landscapes of the type I interferon signaling cascade reveal negative regulation by PJA2

Samira Schiefer [1,2] & Benjamin G. Hale [1] ✉

Deciphering the intricate dynamic events governing type I interferon (IFN) signaling is critical to unravel key regulatory mechanisms in host antiviral defense. Here, we leverage TurboID-based proximity labeling coupled with affinity purification-mass spectrometry to comprehensively map the proximal human proteomes of all seven canonical type I IFN signaling cascade members under basal and IFN-stimulated conditions. This uncovers a network of 103 high-confidence proteins in close proximity to the core members IFNAR1, IFNAR2, JAK1, TYK2, STAT1, STAT2, and IRF9, and validates several known constitutive protein assemblies, while also revealing novel stimulus-dependent and -independent associations between key signaling molecules. Functional screening further identifies PJA2 as a negative regulator of IFN signaling via its E3 ubiquitin ligase activity. Mechanistically, PJA2 interacts with TYK2 and JAK1, promotes their non-degradative ubiquitination, and limits the activating phosphorylation of TYK2 thereby restraining downstream STAT signaling. Our high-resolution proximal protein landscapes provide global insights into the type I IFN signaling network, and serve as a valuable resource for future exploration of its functional complexities.

Virus recognition by infected cells triggers the production of multiple cytokines, including the antiviral type I interferons (IFNs). These type I IFNs (e.g., IFN-α or IFN-β) act by upregulating the expression of antiviral genes (IFN-stimulated genes; ISGs) to restrict virus replication, but they also act to balance the innate immune system and to promote the adaptive immune response[1,2]. IFNα/β receptor 1 (IFNAR1) and IFNAR2 form a heterodimeric complex on the surface of cells to recognize secreted type I IFNs and to signal via the classical Janus kinase/signal transducer and activator of transcription (JAK/STAT) pathway to induce antiviral effects. More precisely, high-affinity binding of type I IFNs to IFNAR2 promotes the dimerization of both receptor chains and permits activation of the receptor-associated intracellular kinases Janus kinase 1 (JAK1) and tyrosine kinase 2 (TYK2)[3,4]. These kinases trans-/auto-phosphorylate, and, in turn, phosphorylate STAT1 and STAT2 on specific tyrosine

residues, leading to STAT1/2 translocation into the nucleus where they assemble with interferon regulatory factor 9 (IRF9) to form the IFN-stimulated gene factor 3 (ISGF3) complex. This transcription factor complex then binds to IFN-stimulated response elements (ISREs) in ISG promoter regions and, with the coordinated action of chromatin-remodeling complexes, activates the expression of many hundreds of ISGs[5–7]. The protein products of ISGs play key roles in limiting virus spread by various potent antiviral mechanisms[2,8].

Dysregulation of the type I IFN signaling cascade is associated with a variety of pathological conditions in humans. For instance, autoimmune diseases (such as systemic lupus erythematosus) can be caused by IFN system overactivation[9–11], while increased susceptibility to severe viral diseases or chronic viral infections can be caused by IFN defects[12]. Thus, type I IFN signaling has to be tightly and intricately regulated with multiple redundant mechanisms,

[1]Institute of Medical Virology, University of Zurich, 8057 Zurich, Switzerland. [2]Life Science Zurich Graduate School, ETH and University of Zurich, 8057 Zurich, Switzerland. ✉e-mail: hale.ben@virology.uzh.ch

which together help to limit potential overactivation while still being able to react rapidly and effectively against invading viral pathogens[13]. Although we do not have a full understanding of the complexities of type I IFN signaling network regulation, it is clear that protein post-translational modifications, such as (de-)phosphorylation, ubiquitination, SUMOylation, and acetylation, can be key players[14]. For example, IFNAR1 can be phosphorylated on a degron motif, leading to its ubiquitination, internalization, and lysosomal degradation in order to prevent hyperstimulation[15–17]. In addition, phosphatases have been reported to dephosphorylate and inhibit JAK1 and TYK2[18,19]. Furthermore, the Suppressors Of Cytokine Signaling 1/3 (SOCS1/3) are negative regulatory proteins that inhibit JAK1/TYK2[20,21] by sterically hindering their kinase activities[22] but have also been described to induce their ubiquitin-mediated degradation[23]. Similarly, ubiquitin-specific peptidase 18 (USP18) interacts with IFNAR2/STAT2 and reduces JAK1 phosphorylation and activity[24]. The identification and molecular characterization of cellular factors that regulate type I IFN signaling has, therefore, proven to be invaluable in unraveling the functionality of this system. In the past, such work has also provided key insights into potential therapeutic targets to treat diseases associated with dysregulation of the IFN system, with key examples being approved drugs, including JAK inhibitors[25–27].

Our current knowledge on type I IFN signaling regulation mainly comes from factor-specific protein interaction studies (e.g., yeast two-hybrid[28–30] screens and immunoprecipitations[31]) or from genetic approaches using mutagenesis[32,33], siRNAs[34] or CRISPR/Cas9-based[5] methods. However, conventional biochemical interaction studies can suffer from biases towards the enrichment of very strong interactors and may miss weak, transient, or even cell context-dependent factors. Gene depletion studies also favor the identification of genes non-essential for cell viability at the expense of equally important factors that may have additional critical cellular roles. In this regard, recent advances in proximity labeling technologies have revealed their potential as powerful tools for deciphering the complex protein environments that underpin cellular signaling networks[35,36]. A major benefit is the possibility of uncovering cell context-dependent, low-affinity, and stimulus-dependent transient interactions in minute-scale temporal resolution[37]. While proximity labeling may be less efficient in identifying low molecular weight protein interactors compared to conventional affinity-based interaction methods[38], this approach is highly complementary to other tools and can uncover unique biological aspects. We therefore aimed to leverage TurboID-based proximity labeling[39], coupled with affinity purification mass spectrometry (AP-MS), to produce an unprecedented global glimpse into the dynamic proximal proteomes associated with the seven core members of the type I IFN signaling cascade: IFNAR1, IFNAR2, JAK1, TYK2, STAT1, STAT2, and IRF9. By attempting to capture temporal changes in protein interactions following type I IFN stimulation, we sought to unveil novel regulatory mechanisms and transient associations that help to orchestrate type I IFN signaling precision. Here, we present high-resolution landscapes that include 103 high-confidence proteins closely associated with the core type I IFN signaling machinery, revealing both known protein assemblies and novel unappreciated complexes. By integrating results from functional screening approaches, we focused on dissecting the role of PJA2 as a previously unrecognized negative regulator of IFN signaling. The E3 ubiquitin ligase PJA2 was found to promote the non-lysine and non-degradative ubiquitination of TYK2 and limit the activating phosphorylation of TYK2, thereby restraining downstream STAT signaling. Thus, our study offers a comprehensive exploration of the individual proximal proteomes associated with the type I IFN signaling cascade and provides an enriched framework to expand our current understanding of this pathway.

## Results

### System-wide identification of the type I IFN signaling proximal proteome

Rapid promiscuous biotinylation of proximal (~35 nm) proteins within minutes makes TurboID an ideal enzyme to study cell signaling events involving low affinity or temporally regulated transient interactions[39,40]. We, therefore, sought to apply TurboID-based proximity labeling and AP-MS analysis to identify proteins in proximity to each of the seven canonical type I IFN signaling cascade components during IFN-α2 stimulation. Using lentiviral vectors, we engineered eleven independent hTERT-immortalized human lung fibroblast (MRC-5/hTERT) cell lines to stably express TurboID-tagged IFNAR1, IFNAR2, JAK1, TYK2, STAT1, STAT2, or IRF9, as well as TurboID-tagged localization controls based on GFP alone or GFP fused to the Lyn11 plasma membrane localization signal[41], a nuclear export signal[42], or a nuclear localization signal[43] (Fig. 1a, Supplementary Fig. 1a). Immunofluorescence microscopy confirmed the homogenous expression and correct intracellular localization of each TurboID-tagged protein (Supplementary Fig. 1b). Furthermore, endogenous IFN signaling component depletion revealed the functionalities of each TurboID-tagged IFN signaling member, with all TurboID-tagged components retaining appreciable activity (Supplementary Fig. 1c). Kinetic analysis of IFN-α2 signaling in MRC-5/hTERT cells revealed that critical events such as STAT1/STAT2 phosphorylation and STAT1 nuclear translocation all occurred between 10 and 120 min post IFN-α2 stimulation (Supplementary Fig. 1d, e), suggesting that timepoints within this timeframe are suitable to identify functionally relevant proteins in proximity to each IFN signaling component. We consequently stimulated each of our engineered cell lines with IFN-α2 at various times (20–120 min) and supplemented the culture medium with exogenous biotin 15 min prior to harvesting in order to biotinylate proximal proteins. After affinity purification of the biotinylated proteins, enriched factors were detected and quantified by label-free mass spectrometry (Fig. 1a). Proteins with a high probability of being proximal to each IFN signaling component at each timepoint post IFN-α2 stimulation were identified by comparing label-free protein quantities against the similarly localized GFP control across three independent replicates with significance analysis of interactome (SAINT)[44,45]. Using stringent selection criteria (4-fold enrichment over the similarly localized GFP control, 2-fold enrichment over all other GFP controls, and an interaction probability of at least 70%), our analyses yielded a total of 103 proteins in proximity to IFNAR1/2, JAK1, TYK2, STAT1/2, and IRF9, which does not include the seven canonical IFN signaling components themselves that were also identified (Fig. 1b, c, Supplementary Data 1). Consistent with previous literature, our data confirmed the constitutive interaction between STAT1 and STAT2[46] and between STAT2 and IRF9[47](Fig. 1d). Additionally, albeit below the interaction probability threshold of 70%, our method highlighted the known transient IFN-α2 stimulated interactions between TYK2 and STAT1/STAT2, STAT1 and IRF9[48,49], and JAK1 and IFNAR2[50] (Fig. 1d). In contrast, the known interaction of TYK2 with IFNAR1[51] was less than 2-fold enriched over the control in our analysis, indicating that our stringent approach compromised some sensitivity. Further supporting the overall validity of our identified type I IFN signaling proximal proteome, several of the proximal proteins could be confirmed by immunoblot analysis of independent small-scale proximity labeling experiments: e.g., the proximity of IL6ST with JAK1 and TYK2 (Fig. 1e), DNAJA2 with STAT1 and STAT2 (Fig. 1f), and CNOT1 with IRF9 (Fig. 1g). Thus, our TurboID-based study has generated a robust dataset of >100 proteins with high probability of being in proximity to the seven canonical IFN signaling cascade components. This dataset will be an essential resource for future exploration of functional aspects of this system.

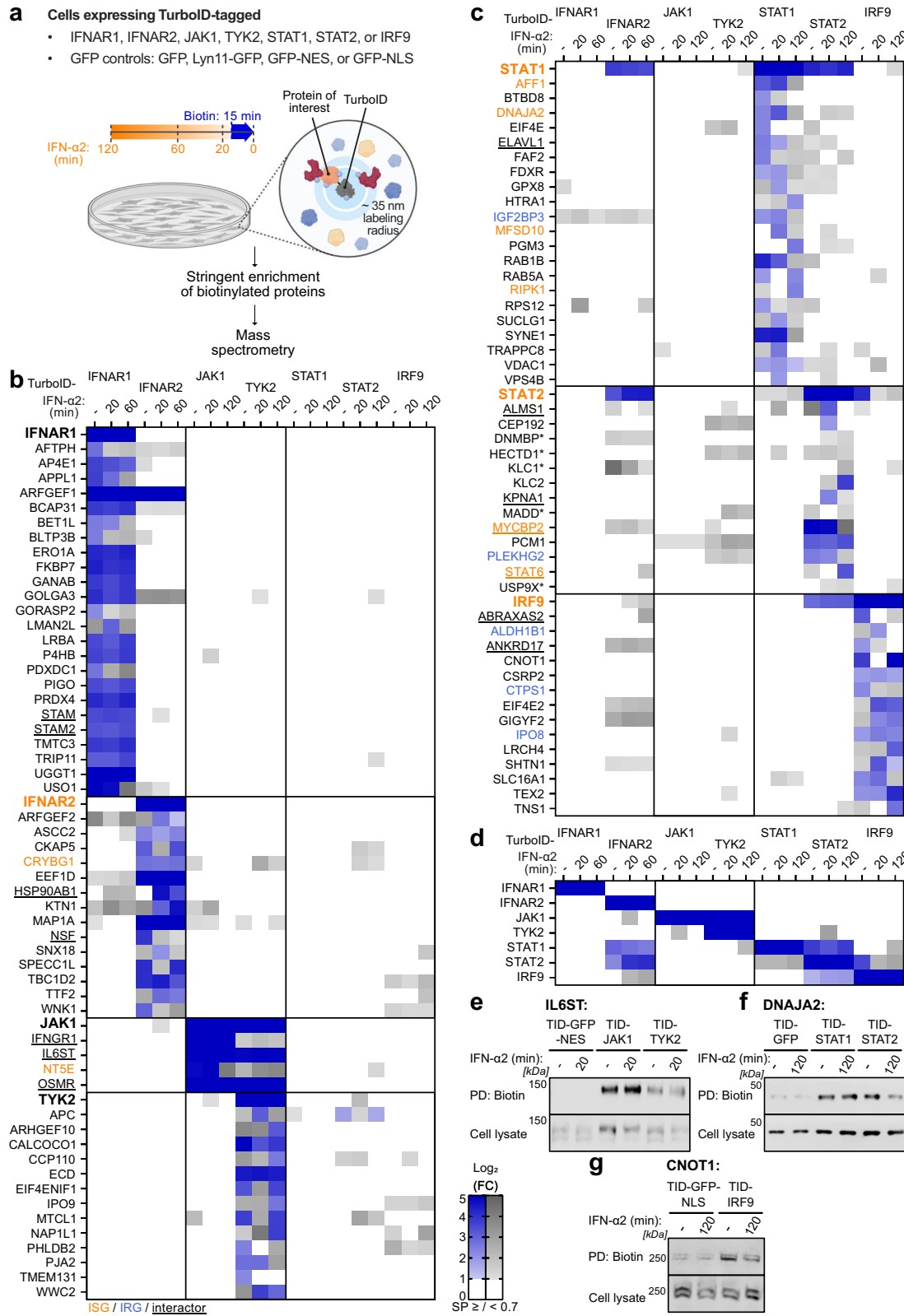

### Global analysis of the type I IFN signaling proximal proteome

Bioinformatic analysis of our type I IFN signaling proximal proteome revealed an interconnected network of factors linked both physically and functionally (Fig. 2). Striking proportions of the factors we identified have previously been associated with components of the type I IFN signaling response (16%, 16 of 103; according to BioGRID[52] and/or STRING[53]). In addition, 13% (13 of 103) are known to be involved in viral processes[54,55], and 8% (8 of 103) are annotated as ISGs[56,57], while 5% (5 of 103) are IFN-repressed genes (IRGs)[56,57]. Notably, 71% of the ISGs and IRGs identified in our study are in close proximity to the transcription factor components STAT1, STAT2, and IRF9. Among the 16 proximal proteins that have been previously linked to type I IFN signaling were STAM and STAM2, which we identified in proximity to IFNAR1 and that have very recently been shown to inhibit IFNAR1-associated TYK2

**Fig. 1 | System-wide identification of the type I IFN signaling proximal proteome. a** Schematic representation of the type I IFN signaling TurboID screening strategy. The indicated MRC-5/hTERT cell lines were stimulated with 1000 IU/mL IFN-α2 for 120, 60, 20, or 0 min, and 500 μM biotin was added for the final 15 min prior to harvesting. Biotinylated proteins were purified with streptavidin beads prior to identification and quantification by mass spectrometry. $n = 3$ independent experiments were performed. **b–d** Heat map representations of $\log_2$(fold change, FC) in the abundance of proteins identified for each of the indicated TurboID constructs at each timepoint post IFN-α2 stimulation. Abundance was determined by label-free quantification in $n = 3$ independent experiments and made relative to the relevant background control (i.e., the similarly localized TurboID-GFP construct). Shown are all proteins that were considered significant according to our selection criteria and that were enriched ≥ 4-fold in at least one condition (see Methods). $\log_2$(FC) ≥ 1 are depicted in blue when the predicted interaction probability (SAINT probability, SP) was ≥ 70%, and in gray when < 70%. $\log_2$(FC) ≥ 5 values are the same shade of blue. **b**, **c** Data for all identified proteins. **d** Data focused only on canonical type I IFN signaling members. In **b**, **c**, previously annotated interactors of canonical type I IFN signaling components according to BioGRID[52] and/or STRING[53] are underlined, while previously identified ISGs[56,57] (> 1.5-fold increased with IFN) are highlighted in orange, and previously identified IRGs[56,57] (> 1.5-fold decreased with IFN) are highlighted in blue. Asterisks (*) indicate factors that were only enriched compared to the GFP-NLS control. **e–g** Immunoblot assessment of three newly identified type I IFN signaling proximal proteins from independent small-scale TurboID experiments performed as described in **a**. Total cell lysates and enriched biotinylated proteins (pull-down, PD) from the indicated conditions were immunoblotted for IL6ST (**e**), DNAJA2 (**f**), and CNOT1 (**g**). Representative of $n = 3$ independent experiments. Source data are provided as a Source Data file.

function in unstimulated cells[58]. In addition, ELAVL1, which we found in proximity to STAT1, is an RNA-binding protein that stabilizes the mRNAs of ISGs[59]. Furthermore, ABRAXAS2, which we identified proximal to IRF9, is a subunit of the BRCC36 isopeptidase complex (BRISC), and is important for ensuring proper STAT1 phosphorylation in response to herpes simplex virus infection[60]. Another factor we identified proximal to IRF9, GIGYF2, appears to be a repressor of ISG activation in the absence of IFN[33].

We additionally analyzed our data with a more relaxed stringency threshold (2-fold enrichment over the similarly localized GFP control) to obtain a greater understanding of potential functional modules and complexes that might be unveiled by our proximal proteome. This relaxed analysis identified an additional 481 proteins in proximity to the seven canonical IFN signaling cascade components (Supplementary Data 2). While our more stringent fold-change (FC) ≥ 4 analysis identified a greater proportion of known IFN signaling interactors according to BioGRID[52] and/or STRING[53], both analyses detected similar percentages of ISGs/IRGs (Fig. 3a, b). As the FC ≥ 2 analysis identified a higher absolute quantity of proximal proteins, it also captured a greater total amount of previously known ISGs/IRGs and interactors. Gene ontology[54,55] analysis of both datasets using DAVID[61] revealed significant enrichment in factors associated with several plausibly relevant biological processes for IFN signaling, such as 'endosomal transport', 'protein import into nucleus', 'protein transport' and 'ER to Golgi vesicle-mediated transport' (Fig. 3c). The identified proteins within these biological processes are highly interconnected according to STRING[53] (Fig. 3d–f). Notably, the majority of proteins involved in 'protein transport' were identified in proximity to IFNAR1/2 (Fig. 3e). Among them, STAM, STAM2, and HGS have previously been demonstrated to interact with IFNAR1 and to regulate IFN signaling[58].

## IFN-stimulated changes to the type I IFN signaling proximal proteome

We next focused on detailing the extent to which proteins identified in the vicinity of type I IFN signaling components changed after stimulation with IFN-α2. To do this, we quantified the FC in protein abundance upon IFN-α2-stimulation relative to the non-stimulated control for all proximal proteins that were significantly enriched compared to the similarly localized GFP control. Proteins were regarded as either enhanced or reduced in proximity to type I IFN signaling components if they displayed a greater than 2-fold change in abundance compared to the non-stimulated control. Sixty-nine out of the 584 proteins analyzed, including 18 from the highly stringent FC ≥ 4 analysis, exhibited increased or decreased abundance near a type I IFN signaling component in response to IFN-α2 stimulation (Fig. 4a). None of the interactions of the canonical type I IFN signaling members with one another met these stringent criteria. More specifically, 34 proteins demonstrated increased abundance, while 35 exhibited decreased abundance in proximity to the

canonical type I IFN signaling components after IFN-α2 treatment (Fig. 4a).

We validated the IFN-α2-stimulated enhancement of KLC2 in proximity to STAT2 using immunoblot analysis of an independent small-scale proximity labeling experiment (Fig. 4b), suggesting that the MS fold changes we calculated are, at least to a certain extent, robust and transferable to other protein detection methods. Mechanistically, we speculate that KLC2 may have a potential role in trafficking STAT2, given its known function in binding specific cargoes that have stimulus-dependent cytosol to nucleus signaling activity via microtubules[62]. Moreover, STAT6, identified here as IFN-α2-enhanced in proximity to STAT2, has been reported to form a transcription factor complex with STAT2 upon type I IFN stimulation[63,64]. Another factor, EP400, showed enhancement in proximity to both STAT1 and STAT2 with IFN-α2 treatment. EP400 is a component of the NuA4 histone acetyltransferase complex, known to activate transcription of specific genes[65]. Consistent with our findings, EP400 was previously identified as an interactor of human STAT2 in high-throughput pull-down assays[66,67] and has recently been detected near murine STAT1 and IRF9 using similar proximity labeling methods[47], indicating its potential involvement in the activation of ISGs. In summary, 12% of our total type I IFN component proximal proteome underwent temporal IFN-α2-stimulated changes, suggesting that these proteins may influence the IFN signaling cascade.

## siRNA screening identifies proximal proteins that functionally regulate type I IFN signaling

To prioritize the identified proximal proteins that influence type I IFN signaling, we used siRNA pools to deplete MRC-5/hTERT cells of 50 high-confidence candidates one by one and subsequently assessed IFN-α2-stimulated (4 h and 16 h) antiviral activity against the IFN-sensitive VSV-GFP reporter virus (Fig. 5a). The 50 candidates were selected due to their high fold-enrichment in the original proximity labeling screen or due to their known functional properties. As controls, we depleted IFNAR2 and IRF9 (two positive regulators of type I IFN signaling), as well as USP18 and SOCS3 (two negative regulators of type I IFN signaling). We initially classified factors as important for type I IFN signaling if their depletion increased or decreased IFN-α2-stimulated antiviral activity by at least 2-fold compared to the non-targeting control siRNA (NTC) in two independent screening runs. Using these criteria, we could confirm IFNAR2 and IRF9 as positive regulators of type I IFN signaling and SOCS3 and USP18 as negative regulators of type I IFN signaling (Fig. 5b–d). In addition, depletion of 16 out of the 50 candidates reproducibly increased the sensitivity of VSV-GFP to 4 h and/or 16 h IFN-α2 stimulation, similar to USP18 and SOCS3 depletion, suggesting that they may also act as negative regulators of type I IFN signaling (Fig. 5d, Supplementary Fig. 2, Supplementary Data 3). These regulators included USP9X, which we identified in proximity to STAT2 in our proteomics screen, and that has not previously been associated with type I IFN signaling but has been

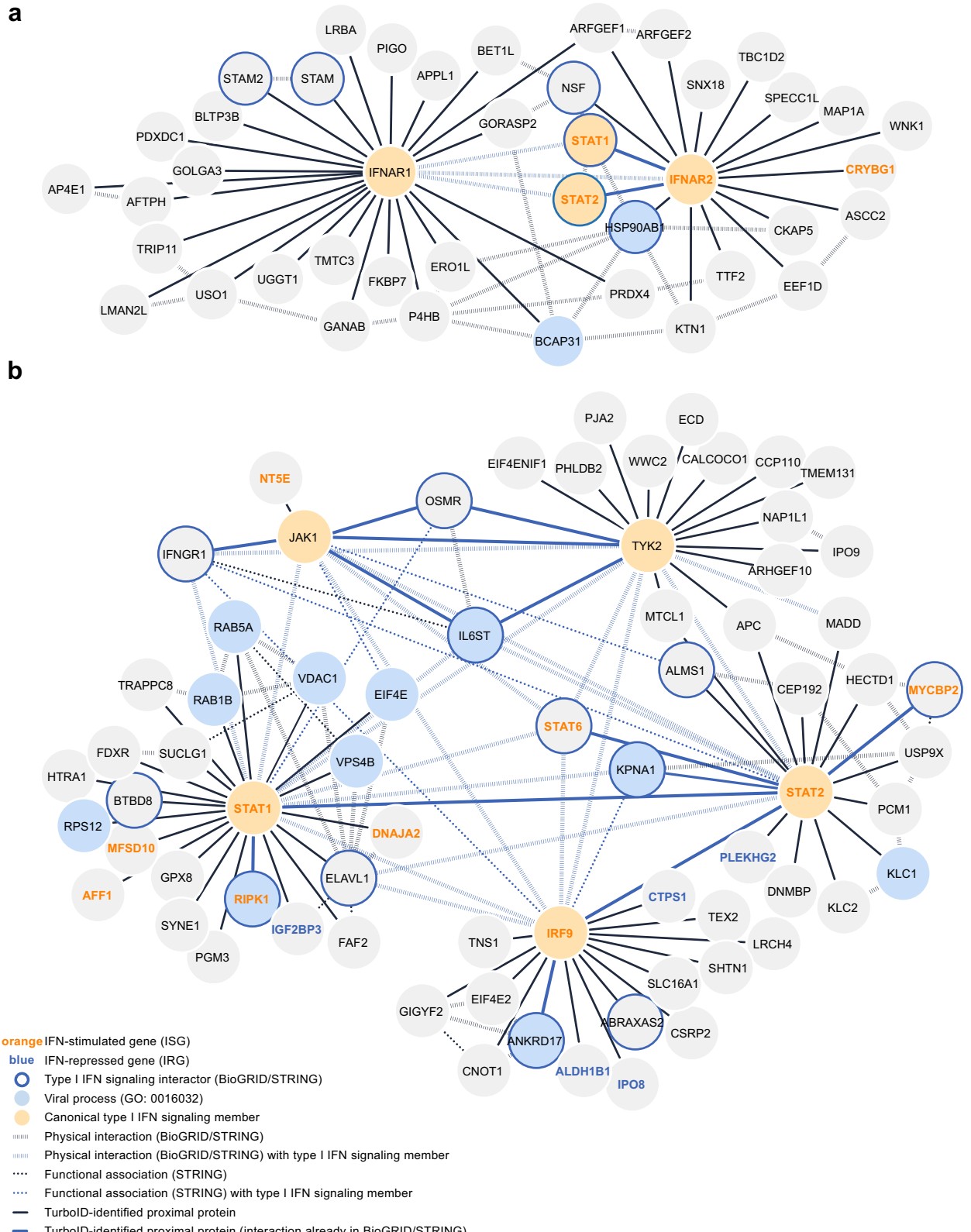

**Fig. 2 | Global analysis of the type I IFN signaling proximal proteome.**
**a** Visualization of the interaction network of proteins identified in proximity to IFNAR1 and IFNAR2. Previously annotated interactions are depicted. **b** Visualization of the interaction network of proteins identified in proximity to JAK1, TYK2, STAT1, STAT2, and IRF9. Previously annotated interactions that were not found in our TurboID screen are depicted where they link proteins found in proximity to the same type I IFN signaling component or toward JAK1, TYK2, STAT1, STAT2, or IRF9. For **a** and **b**, physical interactions and functional associations annotated on BioGRID[52] and/or STRING[53], known ISGs/IRGs, factors involved in viral processes, and the proximal proteins identified in this study are all highlighted as indicated in the figure key. Networks were generated using Cytoscape 3.9[112]. Source data are provided as a Source Data file.

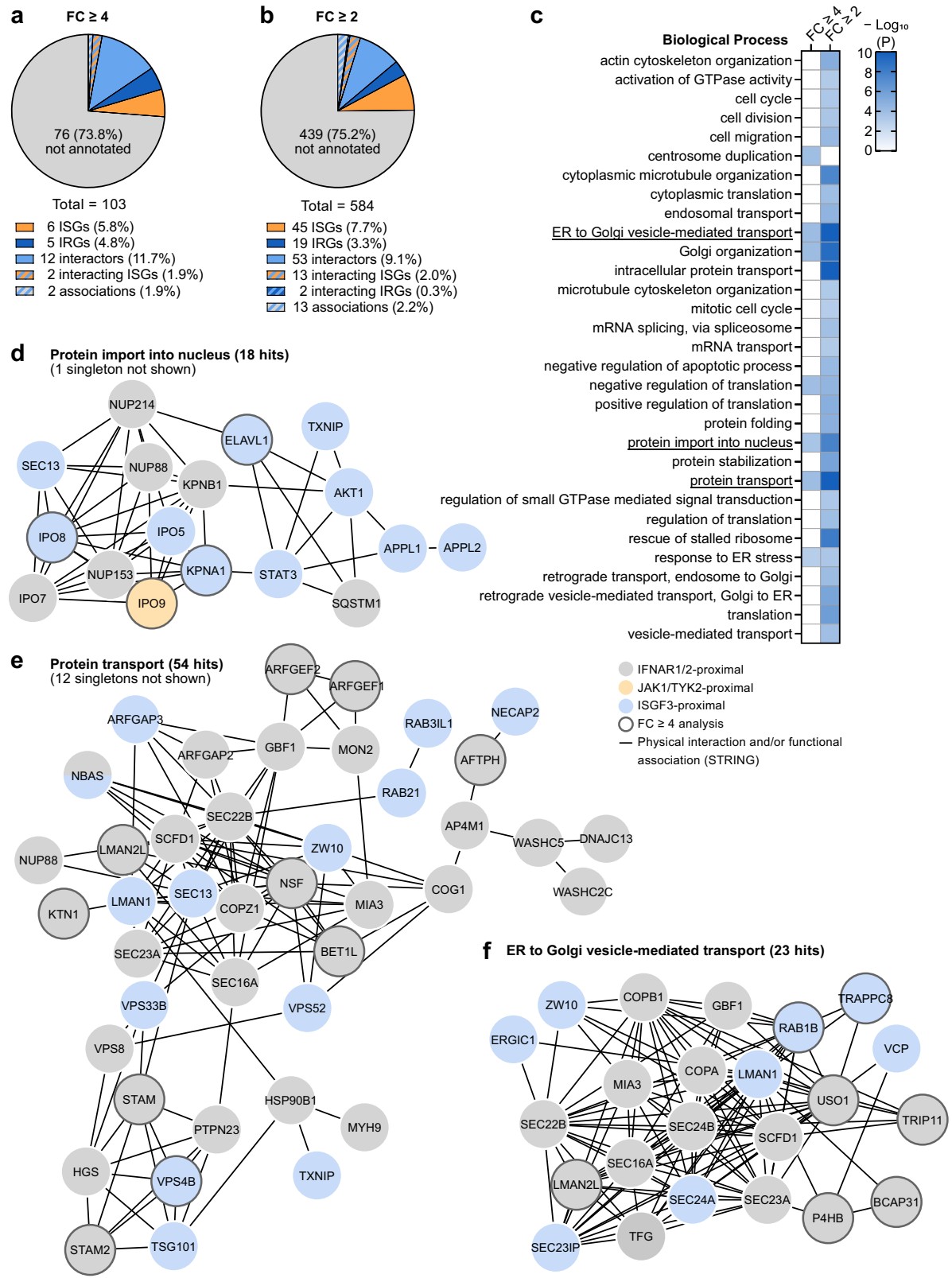

reported to play a role in the closely related JAK2-dependent JAK/STAT signaling pathway[68]. To assess whether any of the identified candidate regulators altered the ability of IFN-α2 to stimulate antiviral gene expression directly, we again used siRNA pools to deplete a subset of the candidates in MRC-5/hTERT cells and assessed IFN-α2-stimulated *ISG54* and *MX1* mRNA expression levels at 4 and 16 h respectively, which were the peak timepoints for individual expression of these two

ISGs. As expected, depletion of IFNAR2 or IRF9 led to clear decreases in IFN-α2-stimulated expression of these ISGs, while depletion of USP18 and SOCS3 led to increases in their expression (Fig. 5e, f). Out of the candidates tested in this assay, depletion of nine (AFF1, ALMS1, CNOT1, IGF2BP3, NT5E, OSMR, PJA2, TBC1D2, and USP9X) led to a greater than 50% increase in IFN-α2-stimulated expression of *ISG54* or *MX1* mRNAs (Fig. 5e, f), further confirming their role as negative regulators with

**Fig. 3 | Functional enrichment in the type I IFN signaling proximal proteome.** **a, b** Pie chart representations of the identified proximal proteins previously associated with type I IFN signaling for stringent (FC ≥ 4; **a**) or relaxed (FC ≥ 2; **b**) analyses. IFN-stimulated genes (ISGs) and IFN-repressed genes (IRGs) are annotated based on 1.5-fold change thresholds[56,57]. Known physical interactors are categorized according to BioGRID[52] and/or STRING[53] where the identified proximal protein has been reported to interact with at least one canonical type I IFN signaling member: IFNAR1/2, JAK1, TYK2, STAT1/2, or IRF9. Other associations were identified with STRING[53] by co-expression, neighborhood, or co-occurrence with at least one canonical type I IFN signaling member. **c** Heat map representation of -log$_{10}$(P value) of functional enrichments in biological process gene ontology terms for the identified type I IFN signaling proximal proteins analyzed with DAVID[61]. One-sided Fisher Exact P values and false discovery rate (FDR) based on adjusted P values were calculated with DAVID[61]. Only P < 0.05 and FDR < 0.05 terms are depicted, where they contain at least 2% of the total proteins identified. −log$_{10}$(P) ≥ 10 values are the same shade of blue. **d–f** Visualization of the physical and functional STRING[53] network of proteins belonging to the significantly enriched biological process gene ontology terms of the identified type I IFN signaling proximal proteins. Proteins identified in proximity to IFNAR1/2, JAK1/TYK2, or STAT1/STAT2/IRF9 (ISGF3) are highlighted as indicated. Networks were generated using Cytoscape 3.9[112]. Source data, including all P values and details on statistical tests, are provided as a Source Data file.

potential direct effects on type I IFN signaling. Interestingly, of these factors, CNOT1 (which we identified in proximity to IRF9) is part of the CCR4-NOT complex that has previously been reported to negatively regulate type I IFN signaling[69,70]. Our integration of loss- and gain-of-function approaches (e.g., reduced VSV-GFP replication coupled with increased ISG expression) should limit the likelihood of experimental artifacts and should give confidence to the screening outcomes reported here. These results indicate that several members of the type I IFN signaling proximal proteome may play functionally relevant roles in this pathway and provide a framework for future studies to validate and investigate specific factors in more detail.

## PJA2 interacts with TYK2 and JAK1 and negatively regulates type I IFN signaling in an E3 ubiquitin ligase activity-dependent manner

We focused follow-up studies on PJA2, a RING-type (Really Interesting New Gene) E3 ubiquitin ligase that was first identified for its ability to control Protein Kinase A (PKA) stability by ubiquitination[71]. PJA2 has since been described to be involved in other signaling pathways, such as the Hippo cascade or Wnt/β-catenin signaling, and has been reported to act either by inducing target degradation or by adding atypical ubiquitin chain linkages to regulate target function[72–74]. PJA2 currently has no known association with IFN signaling, but we identified it in proximity to TYK2 (Fig. 1b, Supplementary Data 1), and our siRNA-mediated depletion studies identified it as a potential negative regulator of IFN-α2-stimulated gene expression and antiviral activity (Fig. 5d, e, and Supplementary Fig. 2). We first confirmed that these functional effects were not limited to MRC-5 cells by performing similar PJA2 depletion studies in human A549 cells, where we depleted PJA2 with either four individual siRNAs or the pool of four siRNAs used in our initial siRNA screen. We found that all PJA2 depletion strategies (similar to depletion of SOCS3) significantly enhanced IFN-α2-stimulated antiviral activity in this cell line (Fig. 6a, Supplementary Fig. 3a). This effect was also observed in primary human bronchial epithelial cells (HBEpCs), where three out of four PJA2 siRNAs significantly reduced VSV-GFP replication after pre-stimulation with IFN-α2 (Fig. 6b, Supplementary Fig. 3b, c). Similar results were obtained in primary HBEpCs from a second independent donor (Supplementary Fig. 3d, e). Further complementing our virological data, IFN-α2-induced expression of ISG54 mRNA was significantly enhanced in primary HBEpCs after PJA2 depletion with both pooled and individual siRNAs (Fig. 6c). Likewise, bulk lentiCRISPR-mediated PJA2 knockout (KO) in HEK293T cells enhanced IFN-α2-induced ISG54-promoter activity in an ISRE luciferase-based reporter assay (Fig. 6d, e).

We next investigated whether the identified proximity of PJA2 with TYK2 could be due to a physical interaction by performing co-immunoprecipitation experiments with tagged forms of TYK2 and PJA2. A V5-tagged TYK2 construct could specifically precipitate endogenous PJA2 (Fig. 6f), while a Flag-tagged PJA2 construct could specifically precipitate endogenous TYK2 (Fig. 6g). These data reveal that TYK2 and PJA2 can physically associate with one another in the same complex, suggesting a relatively stable interaction had been identified in the TurboID-based screen. Interestingly, we found that the related kinase, JAK1, was additionally able to co-precipitate PJA2 when all 7 tagged type I IFN signaling components were individually tested for their interaction with the E3 ubiquitin ligase (Fig. 6h), revealing the specificity of PJA2 interaction with the Janus kinases TYK2 and JAK1. The co-precipitation of PJA2 with JAK1 appeared to be less efficient than with TYK2, which might explain the lack of PJA2 detection as a JAK1 proximal protein in our original TurboID screen. Nevertheless, the interaction of JAK1 with PJA2 was further confirmed for the endogenous proteins (Fig. 6i).

We sought to determine whether the E3 ubiquitin ligase activity of PJA2 contributes to its negative regulation of type I IFN signaling. In an orthogonal approach to our siRNA-based depletion and KO studies, we found that overexpression of wild-type (WT) Flag-tagged PJA2 led to a dose-dependent reduction in IFN-α2-induced ISRE activity in two separate (ISG54 and Mx1) luciferase-based reporter assays (Fig. 6j, Supplementary Fig. 3f, g). Interestingly, overexpression of a previously described PJA2 RING domain mutant (C634A, C671A that lacks E3 ubiquitin ligase activity[71]; PJA2-rm) was unable to reduce IFN-α2-induced ISRE activity (Fig. 6j, Supplementary Fig. 3f, g), revealing the importance of PJA2 E3 ubiquitin ligase activity for the negative regulation of type I IFN signaling. Given that JAK1 plays an essential role in type I, II, and III IFN signaling, while TYK2 activity is restricted to type I and III IFN signaling[4], we also assessed the impact of PJA2 overexpression on type II IFN-induced gamma-activated site (GAS) luciferase reporter activity. Consistent with the observation that PJA2 can interact with JAK1, overexpression of PJA2-WT led to a dose-dependent reduction in IFN-γ-induced GAS activity (Fig. 6k, Supplementary Fig. 3h). Surprisingly, the catalytically inactive PJA2-rm also displayed a slight reduction of GAS activity, albeit to a much weaker extent than PJA2-WT (Fig. 6k, Supplementary Fig. 3h). Immunoblot analysis confirmed that PJA2-WT and the catalytically inactive PJA2-rm expressed to similar levels in these assays (Supplementary Fig. 3i), suggesting that differences in abundance between PJA2-WT and -rm do not account for observed functional effects. These data reveal the ability of PJA2 to physically interact with the Janus kinases, JAK1 and TYK2, and identify a negative regulatory role for PJA2 during type I and type II IFN signaling processes. The ability of PJA2 to inhibit type I IFN signaling is strongly dependent on its E3 ubiquitin ligase function.

## PJA2 promotes ubiquitination of TYK2 and JAK1

To further understand the molecular mechanism by which PJA2 negatively regulates IFN signaling, we investigated whether PJA2 could promote the ubiquitination of JAK1 and/or TYK2. In cell-based ubiquitination assays, we found that V5-tagged JAK1 and V5-tagged TYK2 were more strongly ubiquitinated in the presence of PJA2-WT than in the presence of PJA2-rm, while a similarly tagged control protein (GST-V5) was not ubiquitinated (Fig. 7a–c). We also verified that untagged TYK2 is more strongly ubiquitinated in the presence of PJA2-WT than in the presence of PJA2-rm to exclude any artefactual effects arising from the use of an N-terminal V5 tag on the Janus kinases (Fig. 7d). These data suggest that PJA2 can indeed promote ubiquitination of JAK1 and TYK2. In support of a possible direct

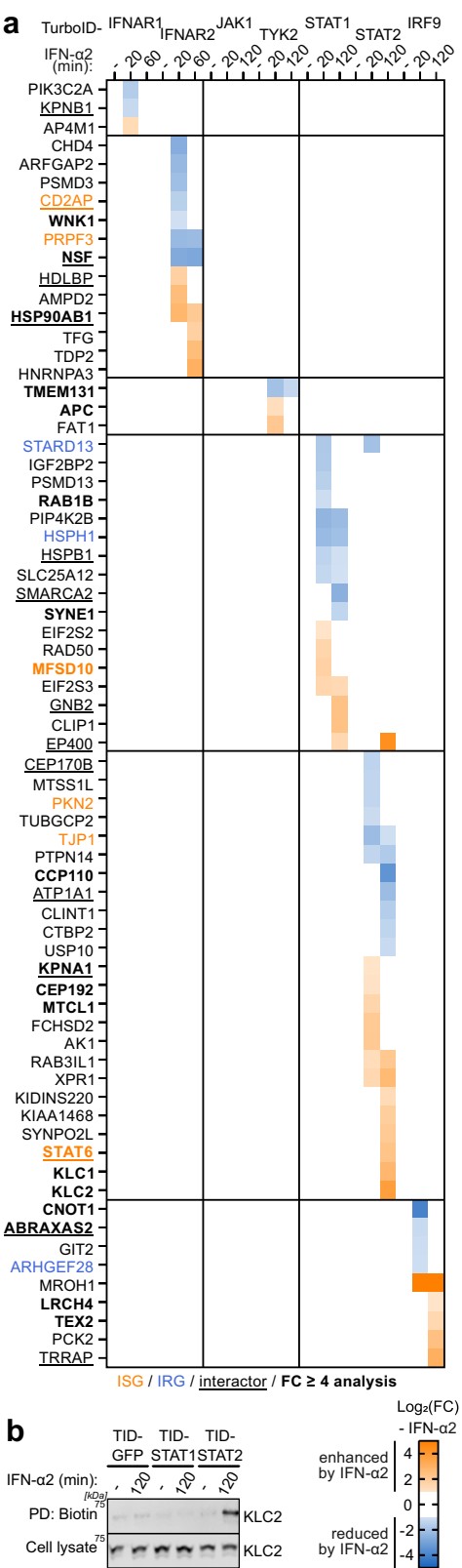

**Fig. 4 | IFN-stimulated changes to the type I IFN signaling proximal proteome.**
**a** Heat map representation of $\log_2$(FC) abundances for interactors of each of the indicated TurboID constructs at each timepoint post IFN-α2 stimulation. Abundance was made relative to the non-stimulated condition. Shown are all proteins that were considered significant according to our selection criteria and that were increased or decreased at least 2-fold compared to the non-stimulated condition (both in FC over the control and in abundance). Only $\log_2$(FC) > 1 and < −1 are depicted. Previously annotated interactors and otherwise associated proteins of canonical type I IFN signaling components according to BioGRID[52] and/or STRING[53] are underlined. Previously identified ISGs[56,57] (> 1.5-fold increased with IFN) are highlighted in orange, and previously identified IRGs[56,57] (> 1.5-fold decreased with IFN) are highlighted in blue. Factors identified as proximal proteins in both analyses (stringent, FC ≥ 4 and relaxed, FC ≥ 2) are highlighted in bold. **b** Immunoblot confirmation of KLC2 in proximity to STAT2 after IFN-α2 stimulation from a small-scale TurboID experiment performed as described in Fig. 1a. Total cell lysates and enriched biotinylated proteins (pull-down, PD) were immunoblotted for KLC2. Representative of $n = 3$ independent experiments. Source data are provided as a Source Data file.

interaction with TYK2 (Fig. 7e, f). This region was previously published to be required for its interaction with the regulatory subunit of another kinase, PKA[71].

TYK2 and JAK1 contain four domains: the N-terminal FERM and SH2 domains, which are both required for their interaction with IFNAR1/IFNAR2, and the C-terminal pseudo-kinase and kinase domains[75–77]. The TYK2 N-terminal FERM and SH2 domains together (residues 1–588) were sufficient to interact with PJA2 in truncation-based mapping studies (Fig. 7g, h). Furthermore, enhanced ubiquitination of this TYK2(1–588) construct, as well as the similar JAK1(1–582) construct, could occur in the presence of PJA2-WT (Fig. 7i, Supplementary Fig. 4b), suggesting that these domains are the predominant ubiquitination targets for PJA2. Canonical ubiquitination occurs on lysine residues[78,79]. To start mapping the ubiquitination site(s) on TYK2, we initially designed a TYK2(1–588) construct that completely lacked any lysine residues (including in the V5-tag) by substituting all 19 TYK2 lysines (as well as 1 lysine in the V5 tag) with arginine residues. Surprisingly, this lysine-less truncated TYK2 construct still exhibited enhanced ubiquitination in the presence of PJA2-WT (Fig. 7j), indicating that the ubiquitination site(s) on TYK2 are likely formed via non-canonical ubiquitination of non-lysine residues, such as serine, threonine, and cysteine; or even to the N-terminal α-amino group of TYK2[80–83]. Additionally, we evaluated the types of polyubiquitin chain linkages PJA2 might conjugate onto TYK2. In polyubiquitin chains, the C-terminus of one ubiquitin molecule is conjugated to another via one of seven lysine residues (K6, K11, K27, K29, K33, K48, K63) or via the N-terminal methionine[84]. Thus, we performed PJA2-dependent TYK2 ubiquitination assays using a panel of specific HA-tagged ubiquitin mutants containing only single lysine residues. While wild-type HA-ubiquitin was readily conjugated to TYK2 in the presence of PJA2-WT, but not PJA2-rm, a complete lysine-less HA-ubiquitin was not conjugated. Furthermore, statistically significant PJA2-dependent ubiquitination of TYK2 in this assay system was observed for the K6-, K27-, K29-, K33-, and K48-only HA-ubiquitin mutants (Supplementary Fig. 4c, d). This suggests that PJA2 can add diverse polyubiquitin chain linkages onto TYK2 and that this effect is not limited to a single homotypic polyubiquitin chain. Overall, these data reveal that PJA2 can promote non-canonical polyubiquitination of TYK2 on the N-terminal FERM and SH2 domains using diverse ubiquitin chain linkages.

## PJA2 promotes non-degradative ubiquitination of the Janus kinases and interferes with TYK2 phosphorylation to limit downstream STAT1 phosphorylation

Ubiquitination is typically thought to induce target protein degradation via the proteasome[78,79]. However, we found that siRNA-mediated

effect, we found that both PJA2-WT and PJA2-rm maintain their interactions with TYK2 and JAK1 (Fig. 7e, Supplementary Fig. 4a), indicating that loss of target ubiquitination by PJA2-rm is not due to impaired target recruitment. Furthermore, using a variety of truncation mutants, we determined that the RING domain of PJA2 is not required for its interaction with TYK2, and instead, PJA2 requires a region spanning amino acids 531 to 630 for strong

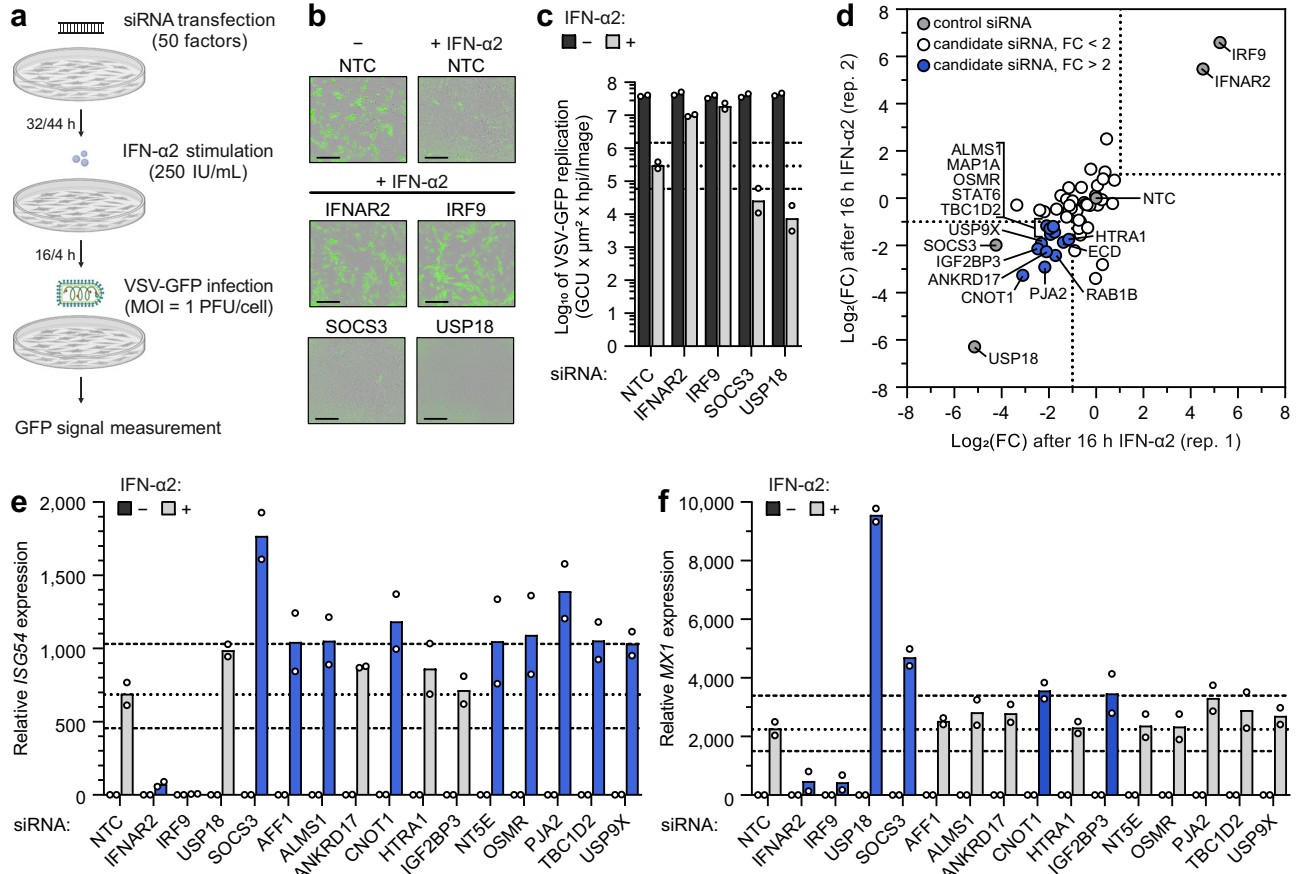

**Fig. 5 | siRNA screening identifies proximal proteins that functionally regulate type I IFN signaling. a** Schematic representation of the VSV-GFP-based siRNA screening approach. siRNA pools targeting 50 factors were individually reverse transfected into MRC-5/hTERT cells prior to stimulation with 250 IU/mL IFN-α2 for 4 or 16 h. Cells were then infected with VSV-GFP (MOI = 1 PFU/cell), and the GFP signal was monitored with the IncuCyte live-cell imaging system as a surrogate readout for viral replication. **b, c** VSV-GFP assay controls showing viral replication in the indicated siRNA-transfected conditions ± 16 h IFN-α2 stimulation. **b** Consists of representative GFP images of infected cells at 48 h post infection. Scale bars represent 300 μm. **c** Total GFP levels calculated from the area under the curve (AUC) values for VSV-GFP replication during the course of the experiment. Bars represent mean values from *n* = 2 independent experiments conducted in technical duplicates. The dotted line indicates VSV-GFP replication in the non-targeting siRNA control (NTC) condition after IFN-α2 stimulation, and the dashed lines

indicate a 5-fold change. **d** Log₂(FC) in VSV-GFP replication (AUC values) after 16 h IFN-α2 stimulation of cells transfected with siRNAs targeting the indicated genes. Log₂(FC) is relative to VSV-GFP replication in the non-targeting siRNA control (NTC) condition. *n* = 2 independent replicates that were conducted in technical duplicates are plotted. The dotted lines indicate a 2-fold change in VSV-GFP replication as compared to the NTC condition. **e, f** *ISG54* (**e**) or *MX1* (**f**) mRNA expression as determined by RT-qPCR following ±4 h (**e**) or ±16 h (**f**) IFN-α2 stimulation (250 IU/mL). Like **a**, MRC-5/hTERT cells had previously been reversing transfected with siRNA pools. Data are normalized to *GAPDH* expression levels in the same sample and made relative to the NTC minus IFN-α2 condition. Bars represent means from *n* = 2 independent experiments conducted in technical duplicates. The dotted line indicates ISG expression in the NTC condition after IFN-α2 stimulation, and the dashed lines indicate a 1.5-fold change from the NTC. Bars with a > 1.5-fold change are colored in blue. Source data are provided as a Source Data file.

depletion of PJA2 did not increase JAK1 or TYK2 protein levels following IFN-α2 stimulation (Fig. 8a, Supplementary Fig. 5a, b). Similarly, overexpression of PJA2 did not reduce JAK1 or TYK2 protein levels following IFN-α2 stimulation (Fig. 8b). We, therefore, reasoned that the ubiquitination of Janus kinases promoted by PJA2 does not lead to target protein degradation, which is consistent with previous studies on PJA2 forming non-degradative atypical polyubiquitin chains on some targets[74,85]. To understand mechanistically how PJA2 may negatively regulate IFN signaling in the absence of TYK2/JAK1 degradation, we next assessed the impact of PJA2 on critical functions of these Janus kinases: regulation of IFN-α/β receptor levels[86] and downstream STAT1 phosphorylation following IFN stimulation[87–89]. More precisely, TYK2 has previously been shown to stabilize IFNAR1 on the surface of unstimulated cells by preventing its internalization and turn-over via endocytosis[90]. Using CRISPR/Cas9, we generated several independent HEK293T-based TYK2 KO cell lines and could confirm that lack of TYK2 reduces total levels of IFNAR1 (Fig. 8c, Supplementary Fig. 5c). Furthermore, re-expression of TYK2 could partially restore IFNAR1

protein levels, but the further expression of PJA2-WT was unable to reverse this (Fig. 8c, d), suggesting that the ubiquitination of TYK2 promoted by PJA2 does not interfere with the ability of TYK2 to stabilize IFNAR1 levels. Since the activity of Janus kinases is influenced by their own phosphorylation status[91], we further evaluated the phosphorylation levels of TYK2 and the target STAT1, following IFN-α2 stimulation in the presence or absence of PJA2 (Fig. 8e–j). We found that siRNA-mediated depletion of PJA2 significantly increased TYK2 phosphorylation after 15 min of IFN-α2 stimulation (Fig. 8e, h), suggesting that PJA2 restrains phosphorylation-mediated activation of TYK2. Indeed, we observed that PJA2 depletion increased the early IFN-α2-stimulated phosphorylation of the target STAT1 in both MRC-5/hTERT and A549 cells (Fig. 8f, g, i, j). The effects were statistically significant at very early times post IFN-α2 stimulation (Fig. 8i, j). Overall, these data are consistent with a working model whereby PJA2 interacts with TYK2 and promotes its non-lysine and non-degradative ubiquitination to impede its activation by phosphorylation, subsequently restraining downstream signaling via the STATs (Fig. 8k).

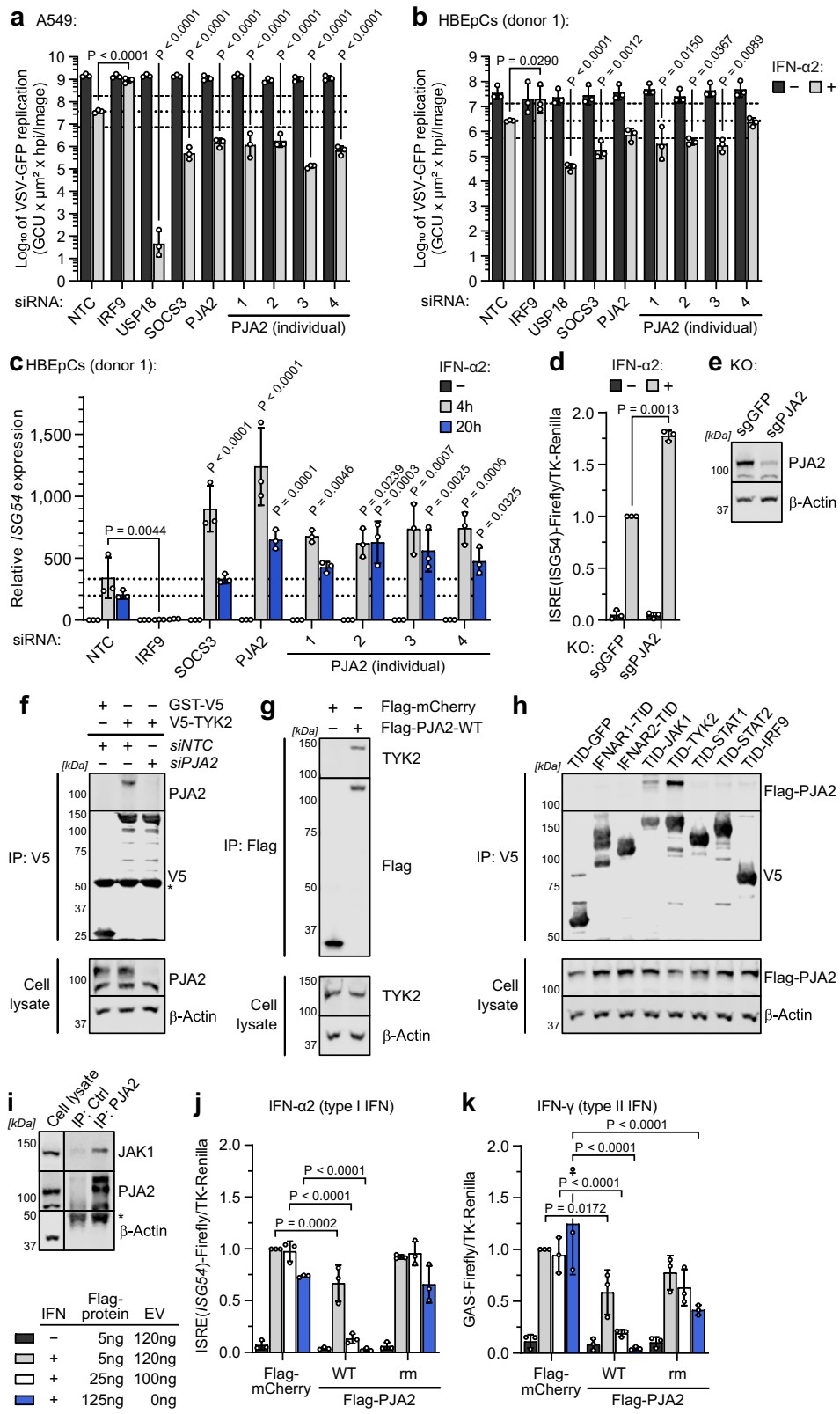

## Discussion

Type I IFN signaling is an essential component of innate defenses against viral pathogens. This signaling cascade must be tightly regulated to prevent aberrant hyperactivation yet still be able to react efficiently to limit viral disease. While key regulators of IFN signaling have previously been identified with different large-scale interaction or genetic approaches[5,28–34], our current study describes the comprehensive system-wide application of TurboID-based proximity labeling and AP-MS to identify 103 high-confidence proteins (FC ≥ 4; 584 proteins with a relaxed FC ≥ 2 threshold) in close proximity to all seven core members of the human type I IFN signaling cascade: IFNAR1, IFNAR2, JAK1, TYK2, STAT1, STAT2, and IRF9. Importantly, analysis of this proximal proteome unveiled an interconnected physical and functional network of factors and revealed

**Fig. 6 | PJA2 interacts with TYK2 and JAK1 and negatively regulates type I IFN signaling in an E3 ubiquitin ligase activity-dependent manner. a, b** VSV-GFP replication in siRNA-transfected A549s ± 16 h IFN-α2 (100 IU/mL) (**a**), or HBEpCs ± 16 h IFN-α2 (10 IU/mL) (**b**). GFP levels were calculated from AUCs. Means ± SDs from $n = 3$ independent experiments are shown. The dotted line indicates replication in the NTC condition with IFN-α2, and the dashed lines indicate a 5-fold change. $P$ values compared to NTC were determined by two-way ANOVA and Šídák's multiple comparisons. **c** *ISG54* expression in siRNA-transfected HBEpCs determined by RT-qPCR following ± 4 h or ± 20 h IFN-α2 (1000 IU/mL). Data are normalized to *GAPDH* and made relative to the NTC condition. Bars represent means ± SDs from $n = 3$ independent experiments. $P$ values compared to the NTC sample were determined by two-way ANOVA and Dunnett's multiple comparisons. The dotted lines indicate expression in the NTC condition with IFN-α2. **d** Bulk PJA2 knockout (sgPJA2) or control (sgGFP) HEK293Ts were co-transfected with ISRE(*ISG54*)-Firefly and TK-Renilla plasmids. Following 16 h of 1000 IU/mL IFN-α2, luciferase activities were determined and normalized to control. Bars represent mean ± SD values from $n = 3$ independent experiments. $P$ value was determined by a two-sided unpaired $t$-test with Welch's correction. **e** Immunoblot from **d** to show protein expression.

Representative of $n = 3$ independent experiments. **f, g** HEK293Ts were co-transfected with plasmids expressing V5-TYK2 or GST-V5 together with NTC or PJA2-targeting siRNAs (**f**), Flag-mCherry or Flag-PJA2 (**g**), or Flag-PJA2 and each TurboID-V5-tagged factor (**h**) prior to generation of cell lysates and immunoprecipitation (IP). Fractions were analyzed by immunoblotting. Representative of $n = 3$ independent experiments. *IgG heavy chain. **i** HEK293T lysates were subjected to IP with anti-PJA2 or anti-luciferase (Control, Ctrl) antibodies. Fractions were analyzed by immunoblotting. Representative of $n = 3$ independent experiments. *IgG heavy chain. **j, k** HEK293Ts were co-transfected with ISRE(*ISG54*)-Firefly (**j**) or GAS-Firefly (**k**) plasmids together with TK-Renilla and 5, 25, or 125 ng of plasmid expressing PJA2-WT, PJA2-rm or mCherry. Total DNA was kept constant with an additional empty vector (EV). Following 16 h of 100 IU/mL IFN-α2 (**j**), or 1000 IU/mL IFN-γ (**k**), luciferase activities were determined and normalized to the 5 ng mCherry condition. Bars represent mean ± SD values from $n = 3$ independent experiments. $P$ values were determined by two-way ANOVA and Dunnett's multiple comparisons. Source data, including all $P$ values and details on statistical tests, are provided as a Source Data file.

previously unappreciated regulatory contributions to the human type I IFN pathway. Our approach is highly complementary to a similar BioID-based proximity labeling screen that was focused on identifying the proximal proteomes of murine STAT1, STAT2, and IRF9 (the ISGF3 complex) in murine macrophage-like cells[47]. While species' and cell type differences could have explained limited overlap in proximal proteins identified in the two screens, it is actually striking that both screens identified 20 common ISGF3-proximal proteins when considering the relaxed FC ≥ 2 threshold for our own dataset (Supplementary Data 4). The commonly identified proteins encompassed factors with known or predicted roles in type I IFN signaling, such as STAT3[92,93], STAT6[63,64], and CNOT1[69,70]. This should give overall confidence to the implementation of proximity labeling for the study of type I IFN signaling. However, the inability to detect a proximal protein does not definitively indicate the lack of interaction, as some proteins are inherently difficult to detect by mass spectrometry based on their size, sequence, or general abundance in the cell type used. Furthermore, proximity labeling relies on the biotinylation of proteins with exposed lysine residues that may be lacking or hidden in a context-dependent manner. Nevertheless, our comprehensive proximity labeling screen performed on all seven core members of the human type I IFN signaling cascade at the same time, which includes high-resolution information on IFN-stimulated interactions of known and unappreciated factors, should serve as a valuable resource for characterizing novel effector mechanisms regulating this signaling cascade.

In our own follow-up studies, we characterized the E3 ubiquitin ligase PJA2 as a negative regulator of IFN signaling. PJA2 interacts with TYK2 and JAK1 and can promote their ubiquitination, strongly inhibiting IFN signaling by a mechanism that appears to limit the activating phosphorylation of TYK2 and consequently restrain downstream STAT1 phosphorylation. Ubiquitination of substrates by PJA2 can lead to their proteasomal degradation[71–73,94]. However, there are also reports of PJA2 ubiquitinating proteins and impacting their functions without inducing degradation. For example, PJA2 can ubiquitinate the HIV-1 Tat protein to stabilize Tat protein levels and activate viral transcription[74], and the PJA2-mediated ubiquitination of MFHAS1 can promote TLR2-mediated signaling[85]. Notably, we found that the ubiquitination of TYK2 promoted by PJA2 is independent of lysine residues in the Janus kinase and does not appear to lead to TYK2/JAK1 degradation. Recently, our knowledge of non-lysine ubiquitination events has greatly advanced, and they are now more broadly recognized as mechanisms to regulate protein functionality[95]. The non-degradative ubiquitination of TYK2/JAK1 could plausibly impact their conformation and/or interaction with other proteins. Indeed, given our observation that IFN-stimulated TYK2 and STAT1 phosphorylation

increase following PJA2 depletion, it is likely that the ubiquitination of JAK1 and TYK2 at least partially impacts their kinase activities. Janus kinases are known to switch between an autoinhibited (closed) and an open conformation in resting cells[96]. Following IFN binding to its receptor, the open conformation is required for Janus kinase dimerization and activation[96]. We hypothesize that PJA2-mediated ubiquitination of TYK2 could stabilize the autoinhibited conformation or act sterically to limit the kinases from dimerizing and TYK2 from being activated by phosphorylation, thereby restraining full signaling activity.

We found PJA2 to be a constitutive interactor of TYK2 and JAK1, which raises the question of whether and how the negative regulatory action of PJA2 itself may get activated. PJA2 does not appear to be an ISG[57], unlike several other negative regulators of type I IFN signaling, such as USP18[57,97] and SOCS1/3[57,98]. A previous study has shown that PJA2 can be activated by serine/threonine phosphorylation in its N-terminal region (S342/T389)[71]. It is possible that an unknown kinase can induce this phosphorylation as part of the IFN signaling response to activate a negative feedback loop. It is further possible that PJA2 acts constitutively to promote low-level ubiquitination of JAK1 and TYK2 to limit hyperactivity of the IFN system and that this inhibitory ubiquitination must, therefore, be overcome to unleash the full signaling potential of the Janus kinases following IFN stimulation.

JAK1 and TYK2 are involved in JAK/STAT signaling pathways other than type I IFN signaling. Prominent examples are the other IFN signaling pathways, type II and type III IFN signaling, as well as IL-2, IL-6, and IL-10 cytokine signaling[99]. Here, we have already shown that PJA2 can affect not only type I, but also type II IFN signaling. Correspondingly, the ubiquitination of Janus kinases promoted by PJA2 might also play a broader role in several different signaling events. Interestingly, we identified one IL-6 family cytokine receptor complex, IL6ST/OSMR, proximal to both JAK1 and TYK2 in our TurboID screen, indicating that we might already have identified additional factors that are involved in other JAK/STAT signaling pathways. This may broaden the applicability of the dataset we provide here.

In summary, our comprehensive TurboID-based study has generated a robust dataset of >100 proteins with a high probability of being in close proximity to the core human components of the type I IFN signaling cascade, including IFNAR1, IFNAR2, JAK1, TYK2, STAT1, STAT2, and IRF9. This dynamic IFN-stimulated map of proteins in the proximity of these key signaling molecules led us to characterize an intriguing mechanism by which non-canonical ubiquitination may fine-tune the IFN pathway and its antiviral action. Our validated dataset should, therefore, form a solid basis for future investigations aimed at unraveling novel functional or regulatory intricacies of type I IFN signaling.

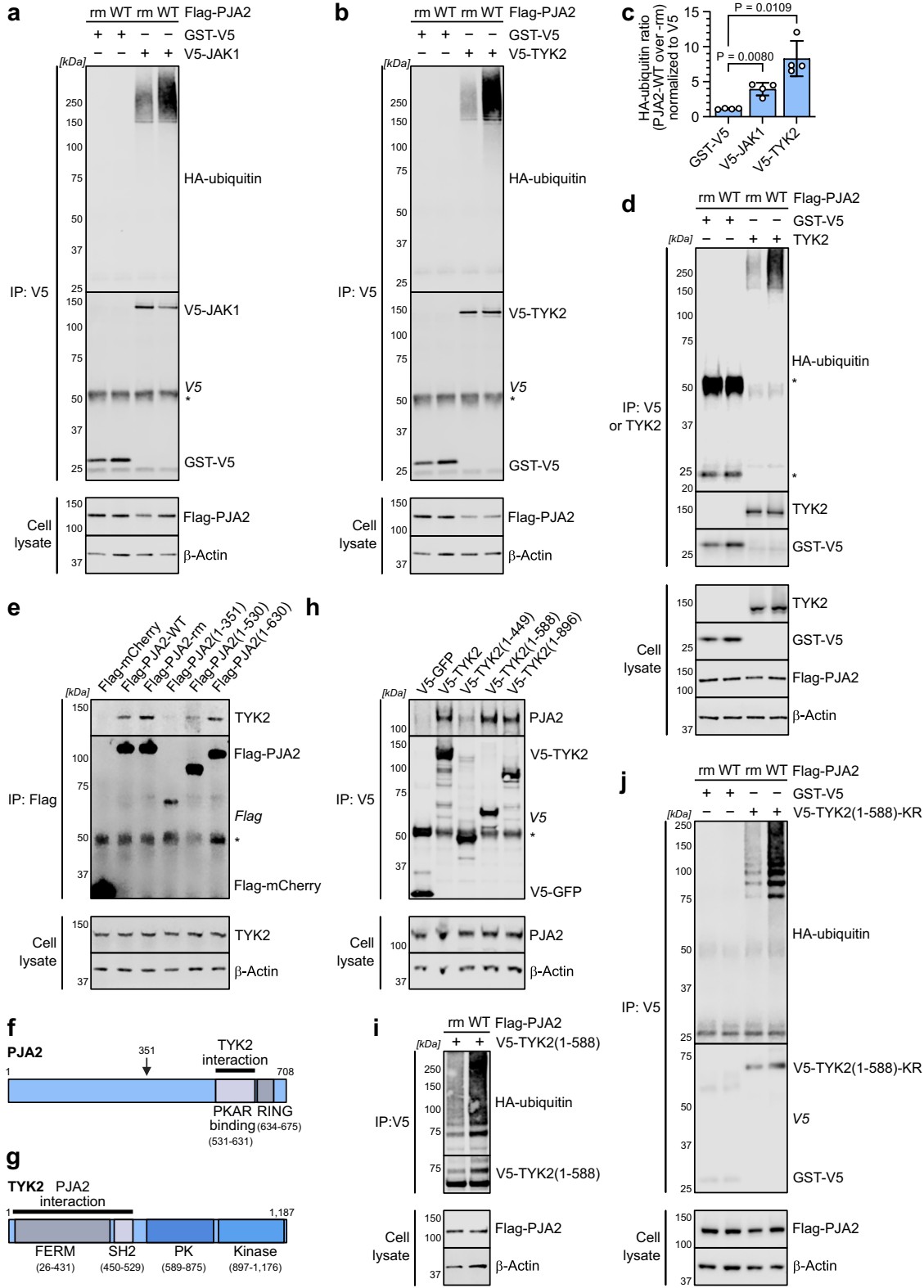

## Methods

### Cell culture

Human embryonic kidney (HEK) 293 T cells and human lung epithelial A549 cells (both originally from ATCC) were cultured in Dulbecco's Modified Eagle's Medium (DMEM; Thermo Fisher Scientific), supplemented with 10% fetal bovine serum (FBS) and 100 U/mL of penicillin-streptomycin (Gibco Life Technologies). hTERT-immortalized human lung fibroblasts (MRC-5/hTERT; a gift from Chris Boutell, University of Glasgow, UK) were cultured in Minimum Essential Medium Eagle (Sigma-Aldrich) supplemented with 10% FBS, 100 U/mL of penicillin–streptomycin, 2 mM GlutaMAX (Thermo Fisher Scientific) and 1% non-essential amino acids (NEAA; Thermo Fisher Scientific). Primary human bronchial epithelial cells (HBEpCs) from a 62-year-old male Hispanic donor (Epithelix; donor 1) and from a 52-year-old female Caucasian donor (PromoCell; donor 2) were cultured in Airway Epithelial Cell Growth Medium (PromoCell) supplemented with the

**Fig. 7 | PJA2 promotes ubiquitination of TYK2 and JAK1. a, b** HEK293T cells were co-transfected with V5-JAK1 or GST-V5 (**a**) and V5-TYK2 or GST-V5 (**b**) together with Flag-tagged PJA2-WT or PJA2-rm and HA-ubiquitin. Cells were lysed in a denaturing buffer containing 2% SDS, which was diluted to 0.7% SDS prior to anti-V5 IP. Cell lysate and IP fractions were analyzed by SDS-PAGE and immunoblotting for the indicated proteins. Data are representative of $n = 4$ independent experiments. **c** Quantification of the HA-ubiquitin signal normalized to the V5 signal of the same IP from the experiments shown in panels (**a**) and (**b**), with HA-ubiquitin/V5 quantities for the PJA2-WT condition made relative to the PJA2-rm condition. Bars represent means ± SDs from the four independent replicates. $P$ values were determined by a two-sided unpaired $t$-test with Welch's correction. **d** Similar experiment to (**b**), albeit an expression vector for untagged TYK2, was used. IPs were performed with anti-V5 or anti-TYK2 antibodies as indicated. Data are representative of $n = 3$ independent experiments. **e, f** HEK293T cells were transfected with plasmids expressing Flag-mCherry or the indicated Flag-PJA2 variants (schematic in **f**) prior to generation of cell lysates and anti-Flag IP. Cell lysate and IP fractions were analyzed by SDS-PAGE and immunoblotting for the indicated proteins. Data are representative of $n = 3$ independent experiments. PKA regulatory subunit binding domain (PKAR binding) and the catalytically active RING domain are indicated in (**f**). **g, h** HEK293T cells were transfected with plasmids expressing V5-GFP or the indicated V5-TYK2 variants (schematic in **g**) prior to the generation of cell lysates and anti-V5 IP. Cell lysate and IP fractions were analyzed by SDS-PAGE and immunoblotting for the indicated proteins. Data are representative of $n = 3$ independent experiments. **i, j** Similar experiments to (**b**), albeit expression vectors for V5-TYK2(1-588) (**i**) or V5-TYK2(1-588)-KR (**j**) were used. Data are representative of $n = 3$ independent experiments. For all panels, * indicates the IgG heavy or light chain visible on the immunoblot of some IP experiments. Source data are provided as a Source Data file.

Growth Medium Supplement Pack (PromoCell) and Y-27632 dihydrochloride (10 µM, Selleck Chemicals). All cells were maintained at 37 °C and 5% $CO_2$. IFN-α2 (NBP2-34971) and IFN-γ (NBP2-34992) were purchased from Novus Biologicals.

## Cloning of plasmids
The cDNA for TurboID[39] was obtained from an existing vector (Addgene plasmid #107169; a gift from Alice Ting, Stanford University, USA) and cloned into pLVX-IRES-Puro (Takara) via BamHI and BglII (V5-TurboID) or MfeI and EcoRI (TurboID-V5). The V5-tag was added using appropriate primers. Full-length *IFNAR1* (NM_174552.2, via EcoRI and BamHI), *IFNAR2* (NM_000874.5, via EcoRI and NotI), and the Lyn11-plasma membrane localizing GFP control (via EcoRI and BamHI) were added to be N-terminal to the V5-TurboID coding sequence. *JAK1* (NM_002227.4, via XhoI and NotI), *TYK2* (NM_003331.5, via EcoRI and NotI), *STAT1* (NM_007315.4, via NotI and BamHI), *STAT2* (NM_005419.4, via EcoRI and NotI), *IRF9* (NM_006084.5, via EcoRI and BamHI), and the other GFP controls (GFP, GFP-NES, GFP-NLS; via EcoRI and BamHI) were cloned to be C-terminal to the TurboID-V5 coding sequence. cDNAs of *STAT1*, *STAT2*, and *GFP* were amplified from pre-existing vectors, and the Lyn11-plasma membrane localization sequence[41] (N-terminal), Rev NES[42] (C-terminal), or SV40 large T antigen NLS[43] (C-terminal) were cloned in-frame with the *GFP* coding sequence using appropriate primers. All other human IFN signaling component cDNAs were obtained by performing RT-PCR using total RNA from MRC-5/hTERT cells as a template. The HA-ubiquitin plasmids were obtained from Addgene (pRK5-HA-Ubiquitin-WT/KR/K48/K63[100], plasmids #17608, #17603, #17605, #17606, gift from Ted Dawson, Johns Hopkins University, USA; pRK5-HA-Ubiquitin-K6/K27/K29[101], plasmids #22900, #22902, #22903, gift from Sandra Weller, UConn Health, USA; pRK5-HA-Ubiquitin-K11[102], plasmid #121152, gift from Josef Kittler, University College London, UK). pRK5-HA-Ubiquitin-K33 was mutated from pRK5-HA-Ubiquitin-KR using the QuikChange XL Site-Directed Mutagenesis Kit (Agilent). The p3xFlag-mCherry-CMV-7.1 and GST-V5 plasmids have been previously described[103,104]. *PJA2* cDNA (NM_014819.5) was obtained from cellular mRNA by RT-PCR and cloned into p3xFlag-CMV-7.1 via BamHI/BglII and NotI. V5-*JAK1*, V5-*TYK2* (via NotI), and untagged *TYK2* (via EcoRI and NotI) were cloned from the pLVX-TurboID-IRES-Puro vectors into pCDNA3.1. pCDNA3.1 containing the *TYK2*(1-588)-KR coding sequence was synthesized by GeneArt gene synthesis (Thermo Fisher Scientific). Specific nucleotide substitutions or coding-sequence truncations were introduced as required using the QuikChange XL Site-Directed Mutagenesis Kit (Agilent). The LentiCRISPR plasmids were cloned from oligonucleotides encoding single-guide RNAs (sgRNAs, Supplementary Table 1). Oligonucleotides were annealed, phosphorylated, and inserted into the BsmBI digested lentiCRISPRv2[105] plasmid (Addgene plasmid #52961; gift from Feng Zhang, Massachusetts Institute of Technology, USA). All plasmid inserts and modifications were authenticated by Sanger sequencing.

## Generation of stable cell lines and lentiCRISPR knockout cell lines
To generate cell lines stably expressing TurboID-tagged proteins or lentiCRISPR knockout cell lines, lentiviral particles were first produced in HEK293T cells by co-transfection of the respective pLVX-TurboID-IRES-Puro construct or lentiCRISPRv2 plasmids, together with psPAX2 and pMD2.G (Addgene plasmids #12259 and #12260; gifts from Didier Trono, EPFL, Switzerland) at a ratio of 2:1:2. After 48 h, lentivirus-containing supernatants were clarified by low-speed centrifugation and filtration through a 0.45 µm filter prior to storage at −80 °C. MRC-5/hTERT or HEK293T cells were subsequently transduced with the lentivirus-containing supernatants in the presence of polybrene (8 µg/mL, Sigma-Aldrich) and were selected with puromycin (2 µg/mL, Thermo Fisher Scientific) for at least 14 days.

## Generation of TYK2 knockout cells
TYK2 knockout cells were generated using a ribonucleoprotein (RNP) system as described previously[106]. Briefly, HEK293T cells were reverse transfected with preassembled RNP complexes using Lipofectamine RNAiMAX transfection reagent (Thermo Fisher Scientific). Knockout cell clones were obtained by limiting dilution and verified via NGS and immunoblotting. crRNA and NGS primer sequences are listed in Supplementary Tables 2 and 3.

## Proximity labeling and protein enrichment
MRC-5/hTERT cells expressing the appropriate TurboID-tagged protein were split 1:4 into 10 cm dishes and left to grow for 5 days until confluent. Cells were then serum-starved for 20 h before treatment with 1000 IU/mL IFN-α2 for 0 min, 20 min, 1 h or 2 h. During the last 15 min, biotin (Sigma-Aldrich) was added to a final concentration of 500 µM. Cells were subsequently washed five times with PBS at 4 °C and lysed in RIPA buffer (50 mM Tris-HCl pH 8, 150 mM NaCl, 0.1% SDS, 0.5% sodium deoxycholate, 1% Triton X-100) supplemented with cOmplete Mini EDTA-free protease inhibitors (Roche). Lysates were sonicated, cleared by centrifugation at $10,000 \times g$ for 15 min, and a fraction was removed for analysis of the total cell lysate. After incubation of the remaining lysate with streptavidin magnetic beads (Thermo Fisher Scientific) for 16 h at 4 °C with end-over-end tumbling, samples were washed twice with RIPA buffer, once with 1 M KCl, once with 0.1 M $Na_2CO_3$, once with 1 M urea in 10 mM Tris-HCl (pH 8.0), and twice in freshly prepared 50 mM ammonium bicarbonate.

## Mass spectrometry
Mass spectrometry (MS) of the proximity labeling experiments was performed by the Functional Genomics Center Zurich. Specifically, the streptavidin magnetic beads were washed and resuspended in digestion buffer (10 mM Tris, 2 mM $CaCl_2$, pH 8.2) prior to on-bead digestion using 10 ng/µL Sequencing Grade Trypsin (Promega) in a microwave instrument (Discover System, CEM) for 30 min at 5 W and 60 °C. The resulting supernatants were transferred into new tubes, and beads

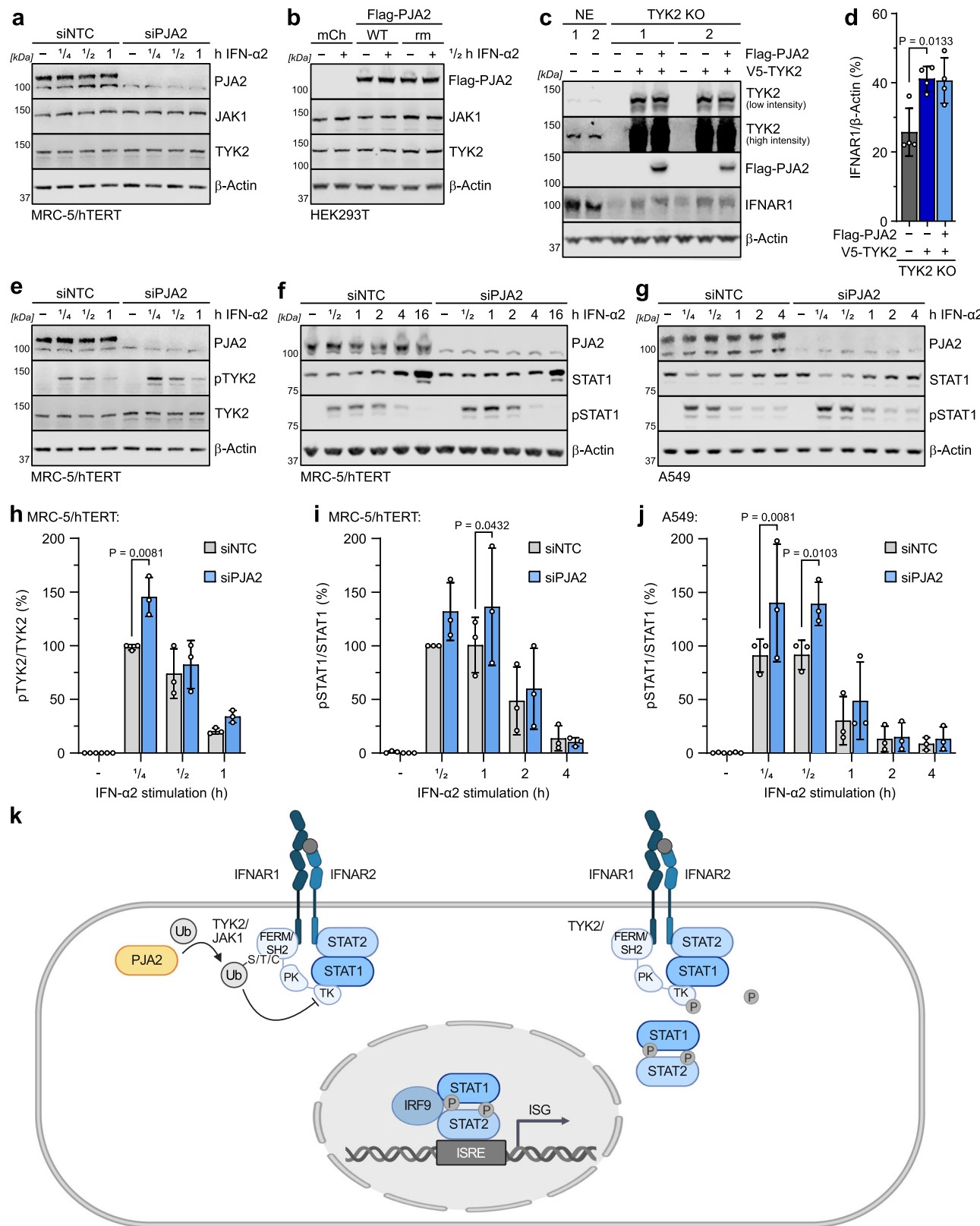

were further digested for 3 h at room temperature before washing with TFA buffer (0.1% TFA, 10 mM Tris, 2 mM CaCl$_2$). Supernatants were collected and pooled with the original before being dried with speed-vac, resolubilized in 0.1% formic acid, centrifuged at 20,000 × $g$ for 10 min, and transferred into LC−MS vials. Mass spectrometry acquisition was performed on a nanoAcquity UPLC (Waters Inc.) connected to a Q Exactive mass spectrometer (Thermo Fisher Scientific) equipped with a Digital PicoView source (New Objective). The solvent composition was 0.1% formic acid for channel A and 0.1% formic acid 99.9% acetonitrile for channel B. Peptides were trapped on a Symmetry C18 trap column (Waters Inc.) and separated on a BEH300 C18 column (Waters Inc.) at a flow rate of 300 nL/min by a gradient (5−35% B in 90 min, 40% B for 5 min and 80% B for 1 min). The mass spectrometer was operated in data-dependent mode, acquiring full-scan MS spectra

**Fig. 8 | PJA2 promotes non-degradative ubiquitination of the Janus kinases and interferes with TYK2 phosphorylation to limit downstream STAT1 phosphorylation. a** MRC-5/hTERT cells were transfected with siRNA pools targeting PJA2 (or NTC) and stimulated with 1000 IU/mL IFN-α2 for the indicated times. Total cell lysates were analyzed by SDS-PAGE and immunoblotting. Data are representative of *n* = 3 independent experiments. **b** HEK293T cells were transfected with the indicated construct and stimulated for 30 min with 1000 IU/mL IFN-α2. Total cell lysates were analyzed by SDS-PAGE and immunoblotting. Data are representative of *n* = 3 independent experiments. **c** NE (non-edited) HEK293T cell clones or TYK2 KO (knock out) cell clones were transfected with the indicated plasmids, and total cell lysates were analyzed by SDS-PAGE and immunoblotting. Data are representative of *n* = 2 similar independent experiments. **d** Quantification of IFNAR1 protein expression normalized to β-actin protein expression and made relative to one NE control clone in replicates of panel (**c**). Bars represent means ± SDs from *n* = 2 independent experiments with *n* = 2 independent NE and *n* = 2

independent KO cell lines (*n* = 4 total). Statistical significance compared to the TYK2 KO cells that were reconstituted with V5-TYK2 was determined by a two-sided unpaired *t*-test with Welch's correction. Non-significant values (*P* > 0.05) are not shown. **e – g** MRC-5/hTERT (**e, f**) or A549 (**g**) cells were transfected with siRNA pools targeting PJA2 (or NTC) and were stimulated with 1000 IU/mL IFN-α2 for the indicated times. Total cell lysates were analyzed by SDS-PAGE and immunoblotting. Data are representative of n = 3 independent experiments. **h – j** Quantification of pTYK2 and pSTAT1 levels normalized to total TYK2/STAT1 protein levels in replicates of panels **e, f** (MRC-5/hTERT, **h, i**) and **g** (A549, **j**). Quantifications were made relative to NTC siRNA-transfected IFN-α2 stimulated samples. Bars represent means ± SDs from *n* = 3 independent experiments. *P* values were determined by two-way ANOVA and Šídák's multiple comparisons. Non-significant values (*P* > 0.05) are not shown. **k** Working hypothesis for how PJA2 acts as a negative regulator of IFN signaling. Source data, including all *P* values and details on statistical tests, are provided as a Source Data file.

(350 to 1500 m/z) at a resolution of 70,000 at 200 m/z after accumulation to a target value of 3,000,000, followed by higher-energy collision dissociation (HCD) fragmentation on the twelve most intense signals per cycle. HCD spectra were acquired at a resolution of 35,000 using a normalized collision energy of 25 and a maximum injection time of 120 ms. The automatic gain control was set to 50,000 ions. Charge state screening was enabled, and single and unassigned charge states were rejected. Only precursors with intensity above 25,400 were selected for MS/MS. Precursor masses previously selected for MS/MS measurement were excluded from further selection for 40 s, and the exclusion window was set at 10 ppm. Three independent replicates of each sample were analyzed in comparison to a similarly localized GFP control using SAINTexpress as previously described[5,45]. In more detail, RAW data was converted into mzML files with MSConvert (version: 3.0.20090-b45f1974b, part of the ProteoWizard software package) using the default settings and a peakPicking filter (parameter: vendor msLevel=1-). Proteins were identified using FragPipe (v13.0) together with the MSFragger Search Engine[107]. Spectra were searched against the canonical Uniprot *Homo sapiens* proteome database (taxonomy 9606, UP000005640 version from 08/2020). Methionine oxidation, N-terminal acetylation, and lysine biotinylation were set as variable modifications, and enzyme specificity was set to trypsin allowing a maximum of two missed cleavage sites. A fragment ion mass tolerance of 0.1 Da and a precursor mass tolerance of 50 ppm were set. SAINTexpress[45] was used to analyze the label-free quantities of *n* = 3 independent replicates for each sample compared to the similarly localized GFP controls with the standard settings applied (lowMode = 0, minFold = 1, normalize = 0). Similarly localized controls were Lyn11-GFP-V5-TurboID (for IFNAR1/2-V5-TurboID), TurboID-V5-GFP-NES (for TurboID-V5-JAK1 and -TYK2), and TurboID-V5-GFP or -GFP-NLS (for TurboID-V5-STAT1, -STAT2, -IRF9). Specific selection criteria were a fold change (FC) ≥ 4 in protein abundance (label-free quantification) over the similarly localized GFP control, as well as an FC ≥ 2 over all other GFP controls, and an interaction probability (SAINT probability) ≥ 0.7 with at least 2 unique spectral counts. For the FC ≥ 2 analysis, selection criteria were an FC ≥ 2 in protein abundance over the similarly localized GFP control and an interaction probability ≥ 0.7 with at least 2 unique spectral counts. Endogenously biotinylated mitochondrial proteins (MCCC1, MCCC2, PC, PCCA, PCCB) were excluded from the analysis. The mass spectrometry proteomics data have been deposited to the ProteomeXchange Consortium via the PRIDE[108] partner repository with the dataset identifier PXD045482.

### Transfections and RNA interference

For plasmid transfections, 70% of confluent HEK293T cells were transfected using FuGENE HD transfection reagent (Promega) according to the manufacturer's protocol. When siRNAs and plasmids were co-transfected into HEK293T cells, reverse transfection with Lipofectamine 2000 (Thermo Fisher Scientific) was performed

according to the manufacturer's protocol. For siRNA-mediated knockdown experiments, MRC-5/hTERT or A549 cells were reverse transfected with ON-TARGETplus siRNA SMARTpools (Horizon Discovery) or FlexiTube siRNA (Qiagen) using Lipofectamine RNAiMAX transfection reagent (Thermo Fisher Scientific) according to the manufacturer's protocol. Detailed information on the siRNAs used in this study can be found in Supplementary Tables 4 and 5.

### VSV-GFP replication assay

MRC-5/hTERT, A549 cells, or HBEpCs were reverse transfected with siRNAs in 96-well plates for 32 h or 44 h before stimulation with 250 IU/mL IFN-α2 (MRC-5) for a further 16 h or 4 h, or with 100 IU/mL IFN-α2 (A549), or 10 IU/mL IFN-α2 (HBEpCs) for 16 h. Cells were then infected with VSV-GFP (MOI = 1 PFU/cell; gift from Peter Palese, Icahn School of Medicine, USA) in FluoroBrite DMEM (MRC-5, Thermo Fisher Scientific) supplemented with 5% FBS or in DMEM (Thermo Fisher Scientific) with 10% FBS (A549) or in Airway Epithelial Cell Growth Medium supplemented with the Growth Medium Supplement Pack (PromoCell) and Y-27632 dihydrochloride (10 μM, Selleck Chemicals; HBEpCs). Cells were monitored with the IncuCyte live-cell imaging system (Sartorius) for up to 2 days. Total green object integrated intensities (Green Calibrated Unit (GCU) × μm²/Image) were exported, and the area under the curve was calculated in GraphPad Prism 9.

### RT-qPCR analysis

RNA was isolated with the ReliaPrep RNA Cell Miniprep kit (Promega) according to the manufacturer's protocol. 250 ng RNA were reverse transcribed into cDNA with SuperScript III or IV (Thermo Fisher Scientific) and an Oligo(dT)$_{15}$ primer (Promega). qPCR was performed with the Fast EvaGreen qPCR Master Mix (Biotium) or PowerTrack™ SYBR Green Master Mix (Thermo Fisher Scientific) and measured on an ABI7300 Real-Time PCR system (Applied Biosystems) in duplicates in 96-well plates. Specific cDNA quantities were calculated based on the ΔΔC$_t$ method and normalized to *GAPDH* as a housekeeping gene. The primers used are listed in Supplementary Table 6.

### Immunoprecipitation (IP)

For standard co-IP experiments, cells were washed once with PBS and lysed in co-IP lysis buffer (50 mM Tris HCl pH 7.5, 150 mM NaCl, 1 mM EDTA, 1% Triton X-100) supplemented with cOmplete Mini EDTA-free protease inhibitors (Roche). Lysates were then sonicated and cleared by centrifugation at 16,000 × *g* for 15 min. For denaturing IP experiments, cells were washed once with PBS and lysed in a denaturing SDS lysis buffer (50 mM Tris-HCl pH 7.8, 650 mM NaCl, 1% NP-40, 2% SDS, 5 mM EDTA, 20 mM N-ethylmaleimide, 10 mM β-mercaptoethanol) supplemented with cOmplete Mini EDTA-free protease inhibitors (Roche). After sonication, lysates were diluted 1:3 in SDS dilution buffer (50 mM Tris-HCl pH 7.8, 650 mM NaCl, 1% NP-40, 0.1% SDS, 5 mM

EDTA, 20 mM N-ethylmaleimide) supplemented with cOmplete Mini EDTA-free protease inhibitors (Roche), and centrifuged at 20,000 × $g$ for 30 min. For all IPs, a sample of the cleared lysate was removed for analysis of the total cell lysate. Cleared lysates were then incubated with the indicated antibodies at 4 °C for 16 h with end-over-end tumbling. Protein G Dynabeads (Thermo Fisher Scientific) were added for 30 min at room temperature with further tumbling. Standard co-IP samples were subsequently washed three times with co-IP lysis buffer while denaturing IP samples were washed six times with SDS dilution buffer. Immunoprecipitated proteins and total cell lysates were analyzed by immunoblotting.

## Immunoblotting

Samples were diluted in 4× Laemmli protein sample buffer (Bio-Rad) containing 10% β-mercaptoethanol or directly lysed in 1× Laemmli buffer and sonicated to shear DNA. Proteins were separated on Bolt 4–12% Bis–Tris Plus Gels (Thermo Fisher Scientific) and transferred onto 0.45 μm nitrocellulose membranes according to the manufacturer's protocol. Membranes were then blocked with 5% milk or 5% BSA in TBS-T, and incubated with the appropriate primary antibody: β-actin (Santa Cruz, sc-47778), actin (Sigma-Aldrich, A2103), Flag (Sigma-Aldrich, F1804 or F7425), V5 (Bio-Rad, MCA1360; or Bethyl Laboratories, A190-119A), GST-HRP conjugate (Cyriva, RPN1236), HA (Cell Signaling Technology, 2367 or 3724), IFNAR1 (Abcam, ab124764), JAK1 (BD Biosciences, 610231), TYK2 (Cell Signaling Technology, 14193), phospho-Y1054/1055-TYK2 (Cell Signaling Technology, 68790), PJA2 (Bethyl Laboratories, A302-991A), STAT1 (Santa Cruz, sc-417), phospho-Y701-STAT1 (Cell Signaling Technology, 9167), STAT2 (Santa Cruz, sc-1668), phospho-Y690-STAT2 (Cell Signaling Technology, 88410), MxA (ab143[109]; gift from Jovan Pavlovic, University of Zurich, Switzerland), IRF9 (BD Biosciences, 610285), DNAJA2 (Abcam, ab157216), KLC2 (Atlas Antibodies, HPA040416), CNOT1 (Cell Signaling Technology, 44613), or IL6ST (Bethyl Laboratories, A304-929A-T). After subsequent washing, membranes were incubated with the appropriate secondary antibody: IRDye 800CW Goat anti-Mouse IgG (LI-COR Biosciences, 926-32210), IRDye 800CW Goat anti-Rabbit IgG (LI-COR Biosciences, 926-32211), IRDye 800CW Donkey anti-Goat IgG (LI-COR Biosciences, 925-32214), IRDye 680RD Goat anti-Mouse IgG (LI-COR Biosciences, 926-68070), IRDye 680RD Goat anti-Rabbit IgG (LI-COR Biosciences, 926-68071), HRP-Horse anti-Mouse IgG (Cell Signaling Technology, 7076P2), HRP-Goat anti-Rabbit IgG (Sigma-Aldrich, A0545), or HRP-Rabbit anti-Goat IgG (Sigma-Aldrich, A4174). The signal was detected with the Odyssey Fc Imager (LI-COR Biosciences) and quantified with the Image Studio Lite Quantification software (LI-COR Biosciences).

## Immunofluorescence microscopy

Cells were seeded on coverslips in 24-well plates for 24 h and, where indicated, were stimulated with 1000 IU/mL IFN-α2. Cells on coverslips were then washed once with PBS, fixed with 3.7% paraformaldehyde for 10 min at room temperature, washed three more times with PBS, and then permeabilized with 100% methanol for 10 min at −20 °C. After three more washes in PBS, samples were blocked in 2% FBS in PBS and incubated with the indicated primary antibodies (see immunoblotting section) in 2% FBS in PBS. DNA was stained with DAPI (Sigma-Aldrich). Samples were stained with the secondary antibodies in 2% FBS in PBS. The secondary antibodies used in this study were Alexa Fluor 488/555 anti-mouse/rabbit (Thermo Fisher Scientific, A11029, A21206, A31570 or A31572). Coverslips were mounted with ProLong Gold Antifade (Thermo Fisher Scientific) and imaged with a Leica DM IL LED microscope (Leica Microsystems) using the LasX software.

## Luciferase reporter assay

In a 96-well plate, HEK293T cells were reverse transfected with a Firefly luciferase-based reporter plasmid (see below), a constitutively active Renilla luciferase control (pRL-TK-Renilla), and various amounts of pCMV7.1–3×Flag-mCherry, -PJA2-WT, or -PJA2-rm. Total plasmid levels were kept constant by the addition of an appropriate amount of empty pCMV7.1–3×Flag vector (EV). Firefly luciferase was expressed either under the control of the murine *Mx1* promoter (pGL3-Mx1P-FFluc[110]; gift from Georg Kochs, University of Freiburg, Germany), the human *ISG54* promoter (pISG54-Luc[111]; gift from Chris Basler, Icahn School of Medicine, USA), or 3 copies of the IFN-γ activation site (GAS; pGAS-Luc[111]; gift from Chris Basler, Icahn School of Medicine, USA). Thirty hours post-transfection, cells were stimulated (or mock) with 100 or 1000 IU/mL of IFN-α2 or 1000 IU/mL of IFN-γ for 16 h. Cells were then lysed, and Firefly and Renilla luciferase activities were determined with the Dual-Luciferase Reporter Assay System (Promega) and an EnVision plate reader (PerkinElmer) according to the manufacturer's protocols.

## Gene ontology analyses

Functional enrichment gene ontology terms[54,55] were analyzed with DAVID[61] using all human genes as background. Bait proteins were excluded from the analyses.

## Network generation

Protein landscape networks were generated using Cytoscape 3.9[112]. Proteins belonging to the viral process (GO:0016032)[54,55] category (Fig. 2) were annotated with Cytoscape 3.9[112]. Protein networks in Fig. 3 were created with STRING[53] and annotated in Cytoscape 3.9[112].

## Statistical analyses

Statistical analyses were performed in GraphPad Prism 9. Data were analyzed by two-way ANOVA and Tukey's multiple comparisons (luciferase reporter assay) Šídák's multiple comparisons (immunoblot quantification of siRNA knockdowns, qPCR quantifications, VSV-GFP assay), or Welch's $t$-test (immunoblot quantification of HA-ubiquitin ratios, IFNAR1 expression levels).

## Reporting summary

Further information on research design is available in the Nature Portfolio Reporting Summary linked to this article.

## Data availability

All data supporting the findings of this study are available within this paper and its supplementary information. The mass spectrometry proteomics data have been deposited to the ProteomeXchange Consortium via the PRIDE[108] partner repository with the dataset identifier PXD045482. Source data are provided with this paper.

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

## Acknowledgements

We are grateful to Davide Eletto (ETHZ, Switzerland) for providing important technical insights at the instigation of this project. We are also grateful to Jovan Pavlovic (University of Zurich, Switzerland), Chris Boutell (University of Glasgow, UK), Alice Ting (Stanford University, USA), Ted Dawson (Johns Hopkins University, USA), Sandra Weller (UConn Health, USA), Josef Kittler (University College London, UK), Feng Zhang (Massachusetts Institute of Technology, USA), Didier Trono (EPFL, Switzerland), Peter Palese (Icahn School of Medicine, USA), Chris Basler (Icahn School of Medicine, USA) and Georg Kochs (University of Freiburg, Germany) for kind gifts of antibodies, cells, plasmids and viruses. Mass spectrometry experiments were performed at the Functional Genomics Center Zurich (FGCZ) of the University of Zurich and the ETH Zurich. Imaging was performed with equipment maintained by the Center for Microscopy and Image Analysis, University of Zurich, Switzerland. Schematic figures (Figs. 1a, 5a, 8k, and Supplementary Fig. 1a) were created with BioRender.com, released under a CC-BY-NC-ND license. The research leading to these results received funding from the Swiss National Science Foundation (grants 31003A_182464 and 310030_214957 to BGH).

## Author contributions

Conceptualization: S.S. and B.G.H.; Methodology, validation, formal analysis, and investigation: S.S.; Visualization: S.S.; Writing—original draft: S.S.; Writing—review & editing: B.G.H.; Supervision: B.G.H.; Funding acquisition and project administration: B.G.H.

## Competing interests

The authors declare no competing interests.
