## [Peer Review File · Nature Communications]

Reviewers' Comments:

Reviewer #1:

Remarks to the Author:

In the manuscript titled "Proximal proteins of the type I interferon signaling cascade," the authors, Samira Schiefer and Benjamin G. Hale, investigate the dynamic interactions within the type I interferon (IFN) signaling pathway. They employed a novel application of TurboID-based proximity labeling coupled with affinity purification-mass spectrometry (AP-MS), to map the human proteomes in close association with key members of the type I IFN signaling cascade under both basal and IFN- α 2 -stimulated conditions. This work led to the identification of a network of 103 proteins proximal to the core signaling components, which included both known and novel protein associations. Through functional screening, the authors identified PJA2 as a negative regulator of IFN signaling, which acts through E3 ubiquitin ligase activity to mediate non-lysine and non-degradative ubiquitination of the Janus kinases, thereby modulating downstream signaling activity.

General comments:

-While the study's approach is innovative, the manuscript would benefit from a comparison with other methods for studying protein-protein interactions, thus contextualizing the advantages and potential limitations of TurboID-based proximity labeling.

-The study relies heavily on mass spectrometry data. The manuscript should discuss any limitations of this technique and how it could affect the interpretation of the results, such as the potential for missing transient interactions or low-abundance proteins.

-The functional assays to validate the role of PJA2 were well-conducted, but the authors might consider additional in vivo studies or clinical data to support the translational relevance of their findings.

-Given the complexity of the IFN signaling cascade, the manuscript would benefit from a discussion on how the identified proximal proteins interact with one another to form functional modules or signaling complexes.

-The potential for off-target effects in the siRNA screening should be addressed, and the use of complementary approaches for gene silencing or knockout could strengthen the claims made.

Specific comments:

-Please submit the actual output file generated from the SAINT analysis. Include at least the high-confidence interactors that have been defined, with average spectral/intensity count, control count, average p-value, SAINT score, BFDR, etc.

-While the work identifies PJA2 as a negative regulator, further investigation is required to fully understand the functional effects of PJA2-mediated inhibition. What is the impact of PJA2's suppression of STAT signaling on cellular responses, such as antiviral defense or immunological modulation? Additional functional assays, such as cell-based assays, could offer a more thorough comprehension.

-The study captures the interactions that occur at specified time intervals after IFN- α 2 stimulation. Gaining insight into the temporal dynamics of these interactions and their evolution over time, particularly in cases of extended exposure to IFN or in chronic infections, could offer a more comprehensive understanding of the regulatory mechanisms.

-It is crucial to explore the clinical consequences of the findings. Is there a dysregulation of PJA2 or its targets in diseases linked to aberrant IFN signaling, such as certain autoimmune disorders or viral infections? An examination of patient samples/data resource and the subsequent correlation of the results with clinical data could determine the significance of these findings in relation to human diseases.

-Line 152-153, "our data confirmed the constitutive interaction between STAT1 and STAT2, and the establishment of a specific STAT2-IRF9 complex lacking STAT1 in unstimulated cells" The authors submitted the reprint file (e.g., reprint_ISGF3.spc.tsv) on the PRIDE website, which clearly demonstrates that the IRF9 protein appears in all three replicates at all three time points (0, 20, and 120 minutes) when either STAT2 or IRF9 is used as bait. Could the author give a justification for their reasoning to get such conclusion?

-Line 194, "siRNA pools to deplete MRC-5 cells of 50 high confidence candidates one by one" Please provide a justification for the selection of these 50 candidate proteins out of 103.

Reviewer #2:

Remarks to the Author:

In this manuscript, the authors took a combined proteomic and genetic approach to identify the negative regulators of type-I interferon (IFN) signaling pathway. Using a Turbo-ID-based proteomic approach, they identified a close IFN signaling component-associated proteins network. These candidates were then followed up using siRNA screen for their antiviral and ISG induction functions. From these screens, they focused on PJA2, which functions as a negative regulator of IFN signaling and its antiviral functions. Mechanistically, PJA2 interacts with JAK1 and TYK2, enhancing their ubiquitination status and inhibiting their functions without causing protein degradation. They also performed domain mapping between PJA2 and TYK2, indicating that PJA2-mediated ubiquitination is independent of lysine. The study is potentially interesting, given the need for negative regulators of IFN signaling, and has some novelty in identifying a new regulator of IFN functions. However, several major weaknesses, both significance of the study and the rigor in approach, dampen the enthusiasm. Some of these points are noted below and may help improve the study.

1. The study focused entirely on human cell lines, and significance may be limited. Does the mechanism apply in mouse cells and/or primary cells?
2. All the interaction studies are done using ectopically expressed epitope-tagged proteins, and some results from endogenous proteins would be required to justify the biological relevance of the study.
3. The screens and interactions studies are powerful; however, their impact on the overall functions of PJA2 is not convincing and is relatively marginal. Results from antiviral (e.g., in Fig 4a) and IFN-signaling (e.g., pSTAT1 in Figs 6f, g) studies may suggest the functions of PJA2 in regulating IFN functions are not strong.
4. Since the function of PJA2 is to ubiquitinate TYK2, using endogenous Ub instead of HA-Ub would produce more meaningful results.
5. It is also unclear why increased ubiquitination of JAK1 and TYK2 would lead to reduced IFN signaling. Are their Ub-defective mutants of JAK1 and TYK2 to address these?
6. What are the types of Ub-linkages PJA2 conjugate on JAK1 and TYK2?
7. What are the immediate consequences of PJA2-mediated ubiquitination of JAK1 and TYK2?

Reviewer #3:

Remarks to the Author:

The study reports an innovative approach to understanding the physical interactions of the local protein effectors of the type I IFN pathway. While there are many studies looking at the mRNA products of the IFN response, this study is important to understanding the local protein-protein interactions that dictate the outcome of IFN binding to its receptor. The study will make a significant contribution to the literature.

The manuscript is well written, easy to read and comprehend. The protein interactome dataset generated should be a valuable starting point for researchers to investigate possible interactors with the 7 major protein effectors of the JAK/STAT dependent type I IFN response. Obvious roles of these interaction partners in the type II and type III IFN responses would also be interesting to investigate.

The introduction includes a lot of unnecessary detail about the mechanism of action of the various regulators of IFN signalling. This level of detail should be condensed to include only the key

important regulators as, in its current form, it is distracting to the reader (Lines 54 to 70). Nicely the data confirms a number of the expected protein-protein interactions expected from IFN binding its receptor – eg: the constitutive interactions of STAT1/STAT2 and IRF9 and validation of other recently described interactions eg STAM and STAM2 with IFNAR1. However, it is curious that the well-defined constitutive interactions between TYK2/IFNAR1 and JAK1/IFNAR2 were not identified. Likewise, the interaction between IFNAR2 and USP18 which is well documented in the literature is also not found in the Mass Spec data set. Can you explain why these interactions are missing from the Mass Spec data? Did you try to immunoprecipitate IFNAR1 and TYK2 (for example) from unstimulated cells (such as you did for IL6ST in Fig 1e)?

Specific comments:

Line 242: JAK1 and TYK2 are also necessary for type III IFN responses alongside type I IFNs. The text should reflect this.

Line 254: The data shows that PJA2 ubiquitinates both JAK1 and TYK2, so why does the title only list TYK2 in this function of PJA2.

Line 286-287: Conclusion that ‘...PJA2 can promote lysine-independent ubiquitination of TYK2, and likely JAK1..’ is speculative. Your data only looked at the lysine-independent ubiquitination of TYK2 and did not address that in relation to JAK1 ubiquitination. The same assumption is made on line 296. These assumption should be corrected to properly reflect the outcomes.

Line 339 ‘... the kinetics transient nature of the ..’ doesn’t make sense.

PJA2 wasn’t identified as binding JAK1 in Fig 1b, yet the immunoprecipitation shows PJA2 binding to JAK1 in the absence of IFN stimulation (Fig 4d). Does this suggest that your TurboID is not sensitive enough to identify this interaction basally? Please explain this discrepancy.

Fig 3c, e and f: RE: statistics with P values displayed on assays that represent only duplicate experiments. Statistically, it’s not very powerful to complete only two replicates. Can a third replicate be completed?

Fig 4: Graphs in Fig 4 e,f and g are really hard to interpret given the way they are presented with P values displayed on all comparisons. Throughout the manuscript, to be consistent between figures and for ease of interpretation by the reader, only show P values on significant comparisons and not on non-significant comparisons. Also it is easier for the reader to interpret significance as asterisks since, depicted as they currently are, the reader has to decide whether a P value is significant especially if all P values are displayed on all comparisons within a figure. In its current state, it is difficult for the reader to identify significant comparisons. Simplify the presentation of graphs.

Also for Fig 4, controls should include 25 ng and 125 ng without IFN treatment, not just 5 ng +/- IFN treatment since transfection can induce an IFN response in the absence of additional exogenous IFN treatment. Luciferase activities should be normalized to each of the 5 ng, 25 ng and 125 ng control conditions. You can’t assume that transfection of additional DNA (ie 25 or 125 ng versus 5 ng) does not alter the response.

Fig 5: Please explain how the data in part c was ‘.. normalized to V5?’

Data in Fig 5j looks very different to that in Fig 5a, b and d with regards to the HA-ubiquitin blot. Please explain why the laddering effect is seen in Fig 5j and not in Fig 5a and b. Also the data in Fig 5j shows binding of the HA-ubiquitin Ab down to products of 75 kDa while those in Fig 5a, b and d are seen only above 150 kDa. Where the experiments carried out under the same conditions? Why do these blots look so different?

Fig 6c: why are 4 lines of NE cells shown on the blot? This is unnecessary.

Part f and g: These blots need to be labelled with the cell type, ie A549s or MRC-5 to be clearly distinguishable. The same for parts h and i. The statistics shown in part h and i comparing siNTC vs siPJA2 – the error bars overlap thus these don’t look significantly different! How was significance determined?

Author response letter for NCOMMS-23-47059

> We thank the reviewers for their positive assessment of our manuscript and for their constructive comments. Please find below all the points raised, with our responses indicated in blue italics.

Reviewer #1:

In the manuscript titled "Proximal proteins of the type I interferon signaling cascade," the authors, Samira Schiefer and Benjamin G. Hale, investigate the dynamic interactions within the type I interferon (IFN) signaling pathway. They employed a novel application of TurboID-based proximity labeling coupled with affinity purification-mass spectrometry (AP-MS), to map the human proteomes in close association with key members of the type I IFN signaling cascade under both basal and IFN- α 2 - stimulated conditions. This work led to the identification of a network of 103 proteins proximal to the core signaling components, which included both known and novel protein associations. Through functional screening, the authors identified PJA2 as a negative regulator of IFN signaling, which acts through E3 ubiquitin ligase activity to mediate non-lysine and non-degradative ubiquitination of the Janus kinases, thereby modulating downstream signaling activity.

General comments:

-While the study's approach is innovative, the manuscript would benefit from a comparison with other methods for studying protein-protein interactions, thus contextualizing the advantages and potential limitations of TurboID-based proximity labeling.

> We have now added a section in the introduction (lines 90-93) highlighting, and citing, a recent study that directly compares proximity labeling with conventional affinity purification, and which discusses the complementary advantages and disadvantages of both methods.

-The study relies heavily on mass spectrometry data. The manuscript should discuss any limitations of this technique and how it could affect the interpretation of the results, such as the potential for missing transient interactions or low-abundance proteins.

> We have now added a limitations section to the discussion noting some caveats of proximity labeling and mass spectrometry approaches (lines 424-428).

-The functional assays to validate the role of PJA2 were well-conducted, but the authors might consider additional in vivo studies or clinical data to support the translational relevance of their findings.

> The reviewer makes a worthwhile point, and it would certainly be of interest in future studies to understand if human genetic variation in PJA2 impacts any immuno- or virological- pathologies in patients, as well as to develop in vivo models of PJA2 function. We believe, however, that such studies are beyond the scope of the present work.

-Given the complexity of the IFN signaling cascade, the manuscript would benefit from a discussion on how the identified proximal proteins interact with one another to form functional modules or signaling complexes.

> This point is well taken. To assess how the identified proximal proteins can form functional modules and to obtain a more wide-spread understanding of this, we chose to additionally analyze our dataset with a lowered, yet still significant, enrichment threshold (2-fold). With the increased number of potential interactors, this allowed us to examine complexes and gene ontology processes in much greater detail, and we have now included a whole new figure (new Fig. 3) and analysis (lines 175-191) on this topic that have greatly improved the revised manuscript.

-The potential for off-target effects in the siRNA screening should be addressed, and the use of complementary approaches for gene silencing or knockout could strengthen the claims made.

> This is a very valid point. We have now added text to the relevant results section (lines 280-288) detailing our opinion that the loss- and gain- of function approaches that we undertook limit the likelihood of such off-target artifacts. Furthermore, to directly address potential off-target effects of the siRNA pool we used for PJA2, we have now performed new additional experiments to complement our original work that utilize either four individual siRNAs targeting PJA2 (new Fig. 6a-c), or newly generated PJA2 knockout cells (new Fig. 6d). These complementary approaches for gene silencing and knockout strengthen the claims we make about PJA2.

Specific comments:

-Please submit the actual output file generated from the SAINT analysis. Include at least the high-confidence interactors that have been defined, with average spectral/intensity count, control count, average p-value, SAINT score, BFDR, etc.

> Thank you for highlighting this oversight. We have now added links to the output files generated from the SAINT analysis as URLs in the Supplementary Data 1.

-While the work identifies PJA2 as a negative regulator, further investigation is required to fully understand the functional effects of PJA2-mediated inhibition. What is the impact of PJA2's suppression of STAT signaling on cellular responses, such as antiviral defense or immunological modulation? Additional functional assays, such as cell-based assays, could offer a more thorough comprehension.

> Thank you for this comment. Figs. 5, 6 and Supplementary Fig. 3 already contain several cell-based assays where we examine the impact of PJA2 depletion on IFN-mediated inhibition of virus replication using VSV-GFP as a model virus. These experiments are also complemented by assays in these same figures looking at the impact of PJA2 depletion on IFN-mediated immune gene expression. We therefore believe that we have addressed the role of PJA2 in antiviral defense and immune-modulation. However, to provide more functional insights into how PJA2 negatively regulates IFN signaling, we have now used further cell-based assays to assess TYK2 phosphorylation in the absence of PJA2, and found that PJA2 depletion significantly increases TYK2 phosphorylation, a marker of TYK2 activation. These new data are included as new panels Fig. 8e, h, and are detailed on lines 391-396.

-The study captures the interactions that occur at specified time intervals after IFN- α 2 stimulation. Gaining insight into the temporal dynamics of these interactions and their evolution over time, particularly in cases of extended exposure to IFN or in chronic infections, could offer a more comprehensive understanding of the regulatory mechanisms.

> The reviewer is right to point out the potential future applications of our newly established experimental system where it would be interesting to study prolonged exposure to IFNs, or the IFN system proximal proteome in the context of chronic (viral) infections. We believe, however, that such studies are beyond the scope of the present work. However, to expand on the reviewer's interest in the temporal dynamics of IFN-stimulated interactions, we have performed further data analyses of how the abundance of proteins in proximity to type I IFN signaling members change over time in response to IFN- α 2 stimulation. This has resulted in new Fig. 4 examined on lines 192-205, which we believe provide added insights into potential regulatory mechanisms to be explored in the future.

-It is crucial to explore the clinical consequences of the findings. Is there a dysregulation of PJA2 or its targets in diseases linked to aberrant IFN signaling, such as certain autoimmune disorders or viral infections? An examination of patient samples/data resource and the subsequent correlation of the results with clinical data could determine the significance of these findings in relation to human diseases.

> The reviewer makes a very worthwhile point that certainly presents an important avenue for future research. However, such an undertaking will require the establishment of appropriate clinical cohorts, and we believe that such efforts are beyond the scope of the present work and are best left to follow up work.

-Line 152-153, "our data confirmed the constitutive interaction between STAT1 and STAT2, and the establishment of a specific STAT2-IRF9 complex lacking STAT1 in unstimulated cells" The authors submitted the reprint file (e.g., reprint_ISGF3.spc.tsv) on the PRIDE website, which clearly demonstrates that the IRF9 protein appears in all three replicates at all three time points (0, 20, and 120 minutes) when either STAT2 or IRF9 is used as bait. Could the author give a justification for their reasoning to get such conclusion?

> *We thank the reviewer for highlighting our original unclear phrasing. We simply wanted to highlight that STAT2 and IRF9 were found in proximity to one another even in the absence of IFN- α 2, while STAT1 was only detected in proximity to IRF9 after IFN- α 2 stimulation. We can see that our original phrasing did not bring that point across properly and we have updated the text on lines 143-147.*

-Line 194, "siRNA pools to deplete MRC-5 cells of 50 high confidence candidates one by one" Please provide a justification for the selection of these 50 candidate proteins out of 103.

> *We have added an explanatory sentence to lines 228-230.*

Reviewer #2:

In this manuscript, the authors took a combined proteomic and genetic approach to identify the negative regulators of type-I interferon (IFN) signaling pathway. Using a Turbo-ID-based proteomic approach, they identified a close IFN signaling component-associated proteins network. These candidates were then followed up using siRNA screen for their antiviral and ISG induction functions. From these screens, they focused on PJA2, which functions as a negative regulator of IFN signaling and its antiviral functions. Mechanistically, PJA2 interacts with JAK1 and TYK2, enhancing their ubiquitination status and inhibiting their functions without causing protein degradation. They also performed domain mapping between PJA2 and TYK2, indicating that PJA2-mediated ubiquitination is independent of lysine. The study is potentially interesting, given the need for negative regulators of IFN signaling, and has some novelty in identifying a new regulator of IFN functions. However, several major weaknesses, both significance of the study and the rigor in approach, dampen the enthusiasm. Some of these points are noted below and may help improve the study.

1. The study focused entirely on human cell lines, and significance may be limited. Does the mechanism apply in mouse cells and/or primary cells?

> *The reviewer makes an important point. We have now expanded our study to assess the impact of PJA2 depletion in primary human cells. We selected primary human bronchial epithelial cells (HBEpC) to investigate this, and used multiple independent siRNAs to show that the impact of PJA2 depletion on ISG expression and viral replication is similar to that in immortalized human cell lines. We performed these experiments in two independent primary cell donors and achieved consistent results. We believe that these new data have enhanced the significance of our study by demonstrating the importance of PJA2 in primary human cells. These new data have been incorporated into Fig. 6b, c, as well as Supplementary Fig. 3b-e. See also lines 280-288.*

2. All the interaction studies are done using ectopically expressed epitope-tagged proteins, and some results from endogenous proteins would be required to justify the biological relevance of the study.

> *We thank the reviewer for highlighting this. To enhance the biological significance, we have now confirmed through co-immunoprecipitation studies that endogenous PJA2 does indeed interact with endogenous JAK1. These new data are presented in Fig. 6i and described on lines 298-302.*

3. The screens and interactions studies are powerful; however, their impact on the overall functions of PJA2 is not convincing and is relatively marginal. Results from antiviral (e.g., in Fig 4a) and IFN-signaling (e.g., pSTAT1 in Figs 6f, g) studies may suggest the functions of PJA2 in regulating IFN functions are not strong.

> *We thank the reviewer for commenting on the strength of our novel screening approaches and the power of our subsequent interaction studies. The reviewer is right to highlight that the overall contribution of PJA2 alone to negative regulation of type I IFN signaling is not absolute, which is of course consistent with the cell possessing multiple negative regulators of this system (such as the*

well-characterized USP18 and SOCS proteins that we use as controls, and which give similar effects to PJA2). As a field we do not yet have a full appreciation of the possible cell-type or context-dependent contributions of each of these negative regulators or their redundancy. Our work should form a basis to explore these possibilities, as it identifies and characterizes a clear and consistent contribution of PJA2 to negative regulation of IFN signaling using multiple orthogonal assay and cell systems: (i) interaction with JAK1/TYK2 in different overexpression and endogenous settings (Fig. 1, Fig. 6); (ii) RING-dependent ubiquitination of JAK1/TYK2 by PJA2 (Fig. 7); (iii) siRNA-mediated depletion of PJA2 enhances IFN-mediated antiviral activity in human cell-lines and primary cells (Fig. 5, Fig. 6, Supplementary Fig. 3); (iv) siRNA-mediated depletion of PJA2 enhances IFN-stimulated gene expression in human cell-lines and primary cells (Fig. 3, Fig. 6); (v) CRISPR-depletion of PJA2 enhances IFN-stimulated gene expression in a human cell-line (Fig. 6); (vi) PJA2 overexpression reduces IFN-stimulated gene expression in a human cell-line in a RING-dependent manner (Fig. 6); (vii) siRNA-mediated depletion of PJA2 enhances IFN-stimulated TYK2 and STAT1 phosphorylation in human cell lines (Fig. 8). We therefore believe that our claim of an impact of PJA2 on IFN signaling is very strong, and rigorously supported by multiple lines of independent experiments and approaches, even if the magnitude of PJA2's impact is smaller than some other essential factors.

4. Since the function of PJA2 is to ubiquitinate TYK2, using endogenous Ub instead of HA-Ub would produce more meaningful results.

> The reviewer makes an important point, and unfortunately our attempts to detect endogenous ubiquitination of TYK2 and JAK1 across various assay conditions have so far been unsuccessful. It is possible that the endogenous ubiquitination levels of endogenous Janus kinases are simply too low to be detected adequately with the assays we have used. In response to this reviewer's point 2 we were, however, able to produce new co-immunoprecipitation data to show that endogenous PJA2 does indeed interact with endogenous JAK1 (Fig. 6i and described on lines 298-302).

5. It is also unclear why increased ubiquitination of JAK1 and TYK2 would lead to reduced IFN signaling. Are their Ub-defective mutants of JAK1 and TYK2 to address these?

> Further investigation is needed to fully understand the direct consequences of JAK1/TYK2 ubiquitination. We had planned to generate specific Ub-defective mutants for precisely the type of investigation proposed by the reviewer, and were surprised that our lysine-deficient TYK2 construct was still ubiquitinated by PJA2 and that our results led us to conclude that ubiquitination occurs on the numerous serines, threonines, cysteines and N-terminus of the protein, and potentially on multiple sites simultaneously. This unfortunately precludes the type of mechanistic study suggested using overexpressed mutants of TYK2/JAK1. However, to overcome this and to provide some initial insights, we took the strategy of depleting cells of endogenous PJA2 and assessing the consequences on downstream IFN signaling. We reasoned that this would produce the similar effect of a non-ubiquitinated JAK1/TYK2 that was specific to the PJA2-targeted residues, would not impact other non-Ub modifications that could target the same residues, and that would function at the fully endogenous level. Using this system, we assessed IFN- α 2 stimulated phosphorylation of TYK2 and STAT1 indicating that the ubiquitination likely limits the phosphorylation and activation of TYK2. These new data are presented in Fig. 8 and described on lines 391-396.

6. What are the types of Ub-linkages PJA2 conjugate on JAK1 and TYK2?

> To address this important point, we have now performed additional TYK2 ubiquitination experiments using single lysine ubiquitin mutants that only contain one lysine and have all other lysine residues substituted by arginine. These mutants can therefore only form one type of homotypic ubiquitin chain linkage. These new experiments have revealed that PJA2 can add various polyubiquitin chains onto TYK2, with no clear single ubiquitin-linkage specificity (polyubiquitination via K6-, K27-, K29-, K33-, and K48 was observed). This implies that PJA2 may not add a simple type of homotypic polyubiquitin chain onto TYK2, but that the ubiquitination is more complex and potentially mixed or branched chained. The new data have been added to the manuscript as Supplementary Fig. 4c, d, and have been described on lines 357-368.

7. What are the immediate consequences of PJA2-mediated ubiquitination of JAK1 and TYK2?

> To address this point, as described in response to this reviewer's point 5, we have performed further PJA2 depletion experiments to mimic loss of JAK1/TYK2 ubiquitination by PJA2 at the endogenous level and studied the consequences for IFN-stimulated TYK2 phosphorylation, a marker of TYK2 activation. We previously found that PJA2-mediated ubiquitination of TYK2 did not lead to TYK2 degradation, and seemed to restrain TYK2 function by limiting STAT1 phosphorylation. In our new experiments, we have found that conditions where PJA2 is present, and can ubiquitinate TYK2, produce an environment where TYK2 phosphorylation is reduced. These new data have been incorporated as new Fig. 8e, h, and support a model whereby PJA2-mediated ubiquitination of TYK2/JAK1 may sterically limit phosphorylation of the tyrosine kinase domain that is required for full Janus kinase activity (updated model Fig. 8k).

Reviewer #3:

The study reports an innovative approach to understanding the physical interactions of the local protein effectors of the type I IFN pathway. While there are many studies looking at the mRNA products of the IFN response, this study is important to understanding the local protein-protein interactions that dictate the outcome of IFN binding to its receptor. The study will make a significant contribution to the literature.

The manuscript is well written, easy to read and comprehend. The protein interactome dataset generated should be a valuable starting point for researchers to investigate possible interactors with the 7 major protein effectors of the JAK/STAT dependent type I IFN response. Obvious roles of these interaction partners in the type II and type III IFN responses would also be interesting to investigate. The introduction includes a lot of unnecessary detail about the mechanism of action of the various regulators of IFN signalling. This level of detail should be condensed to include only the key important regulators as, in its current form, it is distracting to the reader (Lines 54 to 70).

> We thank the reviewer for highlighting this and have now removed some of the detail in this section for clarity.

Nicely the data confirms a number of the expected protein-protein interactions expected from IFN binding its receptor – eg: the constitutive interactions of STAT1/STAT2 and IRF9 and validation of other recently described interactions eg STAM and STAM2 with IFNAR1. However, it is curious that the well-defined constitutive interactions between TYK2/IFNAR1 and JAK1/IFNAR2 were not identified. Likewise, the interaction between IFNAR2 and USP18 which is well documented in the literature is also not found in the Mass Spec data set. Can you explain why these interactions are missing from the Mass Spec data? Did you try to immunoprecipitate IFNAR1 and TYK2 (for example) from unstimulated cells (such as you did for IL6ST in Fig 1e)?

> We thank the reviewer for this comment. As now highlighted in a limitations section of the discussion, there is of course a risk of false negatives within proximity labeling and mass spectrometry experiments due to factors such as lysine availability in target proteins, protein size and the detectability of tryptic peptides. This may account for the missing interactions noted by the reviewer. We have nevertheless delved into our raw data and found that, although below the significance threshold, JAK1 was indeed identified in proximity to IFNAR2 (fold change of 1.2 – 2.2 over the control in proximity to IFNAR2). Moreover, TYK2 exhibited a 1.4 – 1.6-fold enrichment over the control in proximity to IFNAR1. As our minimum arbitrary threshold for analysis was 2-fold with a predicted probability of a true interaction (SAINT probability) ≥ 0.7 , this explains their exclusion (mentioned on line 145-149). Additionally, we likely did not observe the interaction between STAT2 and USP18 as this is normally only observed after prolonged IFN stimulation periods (see: Arimoto, Ki., Löchte, S., Stoner, S. et al. STAT2 is an essential adaptor in USP18-mediated suppression of type I interferon signaling. *Nat Struct Mol Biol* 24, 279–289 (2017). <https://doi.org/10.1038/nsmb.3378>, Ref. 24).

Specific comments:

Line 242: JAK1 and TYK2 are also necessary for type III IFN responses alongside type I IFNs. The text should reflect this.

> We have updated this now.

Line 254: The data shows that PJA2 ubiquitinates both JAK1 and TYK2, so why does the title only list TYK2 in this function of PJA2.

> We originally only listed TYK2 in the title “PJA2 promotes ubiquitination of TYK2 in a lysine-independent manner”, because we did not verify the lysine-independence for the ubiquitination of JAK1. We have now changed the title to reflect that both JAK1 and TYK2 can be ubiquitinated by PJA2.

Line 286-287: Conclusion that ‘...PJA2 can promote lysine-independent ubiquitination of TYK2, and likely JAK1..’ is speculative. Your data only looked at the lysine-independent ubiquitination of TYK2 and did not address that in relation to JAK1 ubiquitination. The same assumption is made on line 296. These assumption should be corrected to properly reflect the outcomes.

> We have now corrected these oversights throughout the text to be more accurate.

Line 339 ‘... the kinetics transient nature of the ..’ doesn’t make sense.

> We have updated this sentence now. An “and” must have been deleted accidentally.

PJA2 wasn’t identified as binding JAK1 in Fig 1b, yet the immunoprecipitation shows PJA2 binding to JAK1 in the absence of IFN stimulation (Fig 4d). Does this suggest that your TurboID is not sensitive enough to identify this interaction basally? Please explain this discrepancy.

> It is indeed possible that TurboID-JAK1 did not biotinylate PJA2 in sufficient amounts to subsequently identify this interaction in our initial proteomics screen. This observation actually correlates well with the immunoprecipitation data (Fig. 6h) where the interaction of PJA2 with JAK1 appears to be less efficient than that of PJA2 with TYK2. We have added a sentence to the main text (lines 298-302) to highlight this point. Of note, however, JAK1 and PJA2 can interact even on an endogenous level in absence of IFN- α 2 (Fig. 6i). We also note that TurboID-JAK1 identified a lower number of proximal proteins (Fig. 1b, c) than the other TurboID constructs, which may indicate fewer interactors or a lower sensitivity specific to the TurboID-JAK1 construct.

Fig 3c, e and f: RE: statistics with P values displayed on assays that represent only duplicate experiments. Statistically, it’s not very powerful to complete only two replicates. Can a third replicate be completed?

> The reviewer rightly comments on the low power of these statistics. However, these two independent screening assays are intended only to provide functional candidates for further mechanistic studies. All of our subsequent work characterizing PJA2 specifically has used 3-4 replicates for determining statistics.

Fig 4: Graphs in Fig 4 e,f and g are really hard to interpret given the way they are presented with P values displayed on all comparisons. Throughout the manuscript, to be consistent between figures and for ease of interpretation by the reader, only show P values on significant comparisons and not on non-significant comparisons. Also it is easier for the reader to interpret significance as asterisks since, depicted as they currently are, the reader has to decide whether a P value is significant especially if all P values are displayed on all comparisons within a figure. In its current state, it is difficult for the reader to identify significant comparisons. Simplify the presentation of graphs.

> We agree that significance is easier to observe when depicted as asterisks vs. actual numerical P values. However, our understanding of the journal guidelines is that exact P values have to be displayed in the figures. However, to make it more consistent and more intuitive, we have now removed the non-significant P values and changed the comparisons to only compare samples with the similarly treated controls.

Also for Fig 4, controls should include 25 ng and 125 ng without IFN treatment, not just 5 ng +/- IFN treatment since transfection can induce an IFN response in the absence of additional exogenous IFN treatment. Luciferase activities should be normalized to each of the 5 ng, 25 ng and 125 ng control conditions. You can’t assume that transfection of additional DNA (ie 25 or 125 ng versus 5 ng) does not alter the response.

> We thank the reviewer for highlighting this lack of clarity on our part. In all these experiments, we had filled up the plasmid amount with empty vector to correspond to the same total amount of DNA for all the different conditions. We have now added such a statement to the figure legend. Furthermore, we had already performed control experiments with the varying amounts of the mCherry control plasmid, and these complete graphs (with all the non-IFN-stimulated samples) are now added as Supplementary Fig. 3f-h.

Fig 5: Please explain how the data in part c was normalized to V5?

> We have now inserted the explanation into the figure legend. The data were normalized to the V5 signal for the same IP sample and then the HA:V5 ratio of the IP from the PJA2-WT sample over the corresponding PJA2-rm sample was calculated.

Data in Fig 5j looks very different to that in Fig 5a, b and d with regards to the HA-ubiquitin blot. Please explain why the laddering effect is seen in Fig 5j and not in Fig 5a and b. Also the data in Fig 5j shows binding of the HA-ubiquitin Ab down to products of 75 kDa while those in Fig 5a, b and d are seen only above 150 kDa. Where the experiments carried out under the same conditions? Why do these blots look so different?

> The reviewer may have inadvertently overlooked this, but in former Fig. 5j (now Fig. 7j) we show a truncated TYK2 mutant that only consists of amino acids 1-588 instead of the full-length protein (1187 amino acids). This truncated V5-TYK2 has a size of 68 kDa instead of the 135 kDa of the full-length V5-TYK2 and is therefore visible between 75-50 kDa in the blot. The lower molecular weight further allows for a better resolution in the gels we used, leading to a clearly visible ubiquitin ladder (Fig. 7j) instead of a smear for the full-length protein (Fig. 7a, b; former Fig. 5a,b).

Fig 6c: why are 4 lines of NE cells shown on the blot? This is unnecessary.

> We have now adjusted the blot to only show two non-edited samples.

Part f and g: These blots need to be labelled with the cell type, ie A549s or MRC-5 to be clearly distinguishable. The same for parts h and i. The statistics shown in part h and i comparing siNTC vs siPJA2 – the error bars overlap thus these don't look significantly different! How was significance determined?

> We have now adjusted the figure to indicate the cell type for each experiment. Significance in former Fig. 6f, g (now Fig. 8h, i) was determined by two-way ANOVA and Šidák's multiple comparisons in GraphPad Prism 9, where similar treated samples from the same replicate were set as 'matched values'. This is how the difference can be significant despite overlapping error bars.

Reviewers' Comments:

Reviewer #1:

Remarks to the Author:

The authors have done a decent job on revising the manuscript and addressed majority of the criticism raised by the reviewers. From my part the manuscript is suitable for publication.

Reviewer #2:

Remarks to the Author:

The authors have addressed my previous concerns.

Reviewer #3:

Remarks to the Author:

The authors have addressed most of my concerns. I am still not convinced that the data in (now) figure 8i is significantly different. Certainly, the data in figure 8h and j show significance and I think that these sections are sufficient to make the point the authors are trying to make. However, I do think the study will make a significant contribution to understanding control of IFN signalling pathways.